# Dynamic Changes in the Gut Microbial Community and Function during Broiler Growth

Maosen Yang,[a,b] Lianzhe Shi,[a] Yile Ge,[a] Dong Leng,[a] Bo Zeng,[a] Tao Wang,[a] Hang Jie,[c] ⓘDiyan Li[a]

[a]School of Pharmacy, Chengdu University, Chengdu, China
[b]College of Animal Science and Technology, Sichuan Agricultural University, Chengdu, China
[c]Chongqing Institute of Medicinal Plant Cultivation, Chongqing, China

Maosen Yang and Lianzhe Shi contributed equally to this work. Author order was determined by discussion and mutual agreement between the two co-first authors.

**ABSTRACT** During the entire growth process, gut microbiota continues to change and has a certain impact on the performance of broilers. Here, we used 16S rRNA gene sequencing to explore the dynamic changes in the fecal bacterial communities and functions in 120 broilers from 4 to 16 weeks of age. We found that the main phyla (*Firmicutes*, *Fusobacteria*, *Proteobacteria*, and *Bacteroides*) accounted for more than 93.5% of the total bacteria in the feces. The alpha diversity of the fecal microbiota showed a downward trend with time, and the beta diversity showed significant differences at various time points. Then, the study on the differences of microbiota between high-weight (HW) and low-weight (LW) broilers showed that there were differences in the diversity and composition of microbiota between high- and low-weight broilers. Furthermore, we identified 22 genera that may be related to the weight change of broilers. The analysis of flora function reveals their changes in metabolism, genetic information processing, and environmental information processing. Finally, combined with microbial function and cecal transcriptome results, we speculated that microorganisms may affect the immune level and energy metabolism level of broilers through their own carbohydrate metabolism and lipid metabolism and then affect body weight (BW). Our results will help to expand our understanding of intestinal microbiota and provide guidance for the production of high-quality broilers.

**IMPORTANCE** The intestinal microbiota has a certain impact on the performance of broilers. However, the change of intestinal microbiota after 4 weeks of age is not clear, and the mechanism of the effect of microorganisms on the weight change of broilers needs more exploration. After 4 weeks of age, the alpha diversity of microorganisms in broiler feces decreased, and the dominant bacteria were *Firmicutes*, *Fusobacteria*, *Proteobacteria*, and *Bacteroides*. There were differences in microbiota diversity and composition between high- and low-weight broilers. Intestinal microorganisms may affect the immune level and energy metabolism level of broilers through their own carbohydrate metabolism and lipid metabolism and then affect the body weight. The results are helpful to increase the understanding of intestinal microbiota and provide reference for the production of high-quality broilers.

**KEYWORDS** broilers, gut, feces, microbiota, body weight, diversity, function

**Ad Hoc Peer Reviewer** ⓘ Stanley Chen, Food Standards Australia New Zealand; ⓘRabindra Mandal, University of Arkansas at Fayetteville; ⓘGajender Aleti, University of California, San Diego

Address correspondence to Hang Jie, jiehangisgood@126.com, or Diyan Li, lidiyan860714@163.com.

The authors declare no conflict of interest.

The gut is an important digestive organ of broilers, and the gut microbiota is also heavily involved in the physiological functions of the gut (1). Some studies shown that gut microorganisms can affect host metabolism (2–6), disease resistance (7, 8), and development. For example, gut microorganisms can help the host uniquely digest and absorb polyphenols, flavonoids, and other substances that are difficult to metabolize in the gut (9–11). Metabolites of gut microorganisms such as short-chain fatty acids also participate as crucial regulators of immune responses to maintain host immune homeostasis (12). In addition, accumulating evidence has shown that the gut microbiota affects the growth and development of the

host. Microbial enzymes have been shown to contribute to differences in intestinal morphology by influencing ileal viscosity and ileal digestibility of starch (13). Further, modulation of gut microorganisms improved the body weight (BW) and bone strength of broilers (14).

As important agricultural animals, broilers play the role of providing high-quality meat. Therefore, gut microorganisms related to body weight (BW) have attracted our attention. In mice, though germfree mice had higher chow consumption than did control mice, their body weight was still lower than that of control mice (15). The study revealed that the gut microbiota is a necessary condition for the body weight gain of the host. Numerous current studies have identified that gut microbiota has the potential to increase the body weight of broilers. *Bacteroidetes* and *Firmicutes* have been widely identified as being able to increase the body weight of the host by affecting the ability to harvest energy from the diet (16). More importantly, a colonization study of the mouse gut microbiota demonstrated that obesity can be transmitted by specific colonization of the gut microbiota (17). Fecal transplantation could also improve the growth performance and liver fat metabolism of broilers by reshaping the intestinal microbiota (18). Meanwhile, several studies demonstrated that the gut microorganisms of broilers in a high-density production environment were significantly different from those in the wild (19, 20). Thus, to fully understand and utilize the effect of the gut microorganisms on the weight gain of broilers, it is necessary to explore the body weight-related gut microbiota.

Previous researches on the dynamic changes in gut microorganisms showed that gut microorganisms were inherited partly from maternal hens before birth and then greatly influenced by the environment after birth (21). A previous study demonstrated that the microbiota tends to increase in diversity during embryonic development and the early stages after birth and develops into a relatively mature community and starts to become relatively stable at 3 to 4 weeks of age (22–24). Referencing previous research, we used amplicon sequencing of the 16S rRNA gene to explore and supplement the role of changes in broiler gut microorganisms after 4 weeks. Research pointed out that fecal samples can be effectively used to detect some shifts and responses in the gut microbiota (25). Therefore, this study focused on changes in fecal microbial diversity and function during growth and further attempted to explore body weight-related fecal microbiota to determine dynamic changes in the gut microbial community and improve the production of broilers.

## RESULTS

**Changes in gut microbial diversity during broiler growth.** In order to comprehensively understand the dynamics of the gut microbial community in broilers during the growth process, a total of 60 fecal samples (10 samples per stage) collected from six time points (4, 5, 8, 9, 14, and 16 weeks) were subjected to the microbial diversity study through a 16S rRNA gene high-throughput sequencing approach. After quality control of the sequence data, a total of 3,296,944 high-quality sequences were obtained with an average of 41,212 reads per sample, and the sample with the lowest sequence number had 26,943 reads. After sequence deredundancy, we constructed the feature metadata table of all samples, which contained 6,405 high-quality features (see Table S1 in the supplemental material). We selected the sequence number of 15,000 for subsequent analysis. In this case, the rarefaction curves were close to saturation, indicating that the sequencing depth was adequate in all the samples (Fig. S1).

Shannon and Faith's phylogenetic diversity (PD) indices were used to represent the alpha diversity of microbiota; the results showed a significant difference between weeks 5 and 16 ($P < 0.05$). And with the growth of broilers, the alpha diversity showed a downward trend (Pearson, $r = -0.74$ and $P = 0.09$) (Fig. 1A and B; Table S3). However, the body weight of broilers changed significantly and increased from the initial average weight of 304.2 g in week 4 to 3,105.8 g in week 16 (Fig. 1C; Table S2). Significant separation of gut microbiota from different time groups was observed using principal-coordinate analysis (PCoA) (Fig. 1D). By using permutational multivariate analysis of variance (PERMANOVA) pairwise comparison, we confirmed that there were significant differences in the beta diversity of gut microbiota except samples of weeks 4 and 5, and weeks 9 and 16 ($P < 0.05$) (Table S4). These results

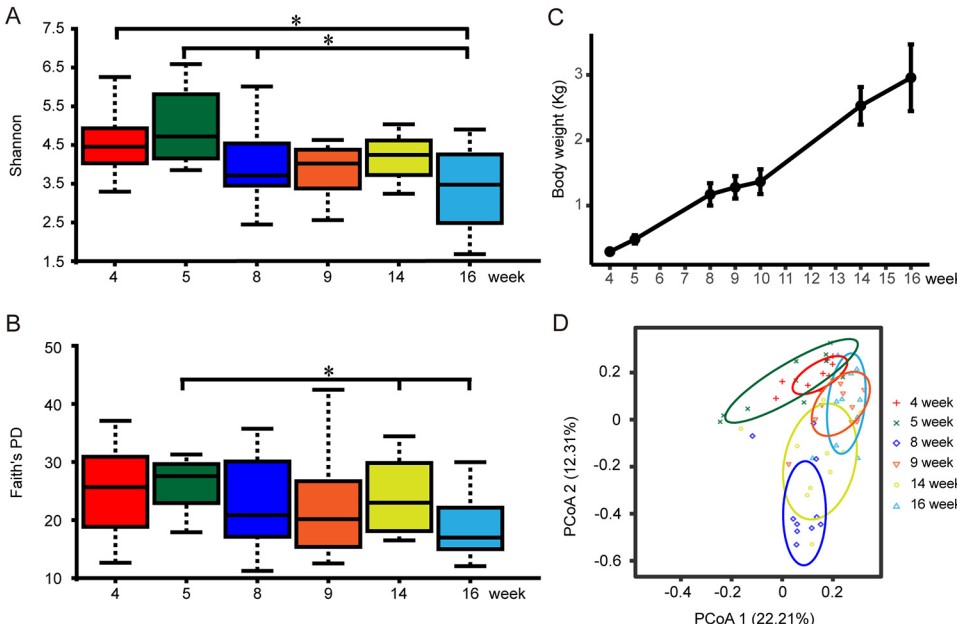

**FIG 1** Changes in microbial diversity and body weight of experimental broilers. (A) The Shannon index at six time points. The box plot shows the quartile, median, and extreme limit of the value. The Kruskal-Wallis H test was used to test the significance of differences. (B) Faith's PD index at six time points. The box plot shows the quartile, median, and extreme limit of the value. The Kruskal-Wallis H test was used to test the significance of differences. (C) The body weight of experimental broilers at seven time points. (D) Principal-coordinate analysis (PCoA) based on weighted Bray-Curtis distance. PERMANOVA was used to detect the significance of separation between groups.

showed that there were significant changes in the gut microbial community of broilers after 4 weeks of age.

**Taxonomic changes and analysis of the dominant bacteria.** In the following taxonomic analysis, we focused on the bacterial community composition at the phylum and genus levels. Compared with the SILVA database (132_99 release), totals of 36 phyla, 89 classes, 216 orders, 390 families, 852 genera, and 1,176 species were identified among 6 groups. The most abundant phyla (top 10 from all samples) and genera (top 20 from all samples) with high average relative abundance were displayed using the stacked bar chart (Fig. 2). At the phylum level, *Firmicutes* (67.35%), *Fusobacteria* (9.85%), *Proteobacteria* (8.36%), and *Bacteroidetes* (8.09%) were the 4 most dominant phyla in fecal samples, dominating the bacterial community by more than 93% (Fig. 2A; Table S5). At the genus level, *Lactobacillus* had the highest relative abundance in weeks 4, 5, 9, and 16 and maintained a high abundance during the entire growth process. Compared with other times, the relative abundance of *Romboutsia* was the highest in week 14 (26.84%). *Fusobacterium* became the most abundant genus (44.25%) in week 8, but it showed lower abundances at other times (0.25% in week 4, 1.31% in week 5, 4.3% in week 9, 7.29% in week 14, and 1.65% in week 16) (Fig. 2B; Table S5).

To elaborate more nuanced changes, we screened the genera whose abundance was significantly related to the sampling time points ($P < 0.05$ and false-discovery rate [FDR] of <0.25). At the phylum level, we observed that the genera with reduced abundance were mainly concentrated in *Firmicutes*, *Proteobacteria*, *Bacteroidetes*, and *Actinobacteria*. Moreover, the genera with increased abundance were mainly concentrated in *Firmicutes* and *Proteobacteria* (Fig. 3A). We used linear discriminant analysis (LDA) effect size (LEfSe) to compare bacterial relative abundances at the genus level. Based on LEfSe results, we observed a total of 8 significantly different genera from weeks 4, 14, and 16 (LDA > 4 and $P < 0.05$) (Fig. 3B). There were 4 significantly different genera in week 14, indicating that the gut microbiota continues to change even in week 14. These results indicated that the gut flora changes dynamically during the growth of broilers.

**Analysis of gut microbial diversity and composition in high- and low-weight broilers.** To obtain a more precise understanding of the differences in gut microbial diversity between high- and low-weight broilers, the five samples with the highest body weight in week 9 were

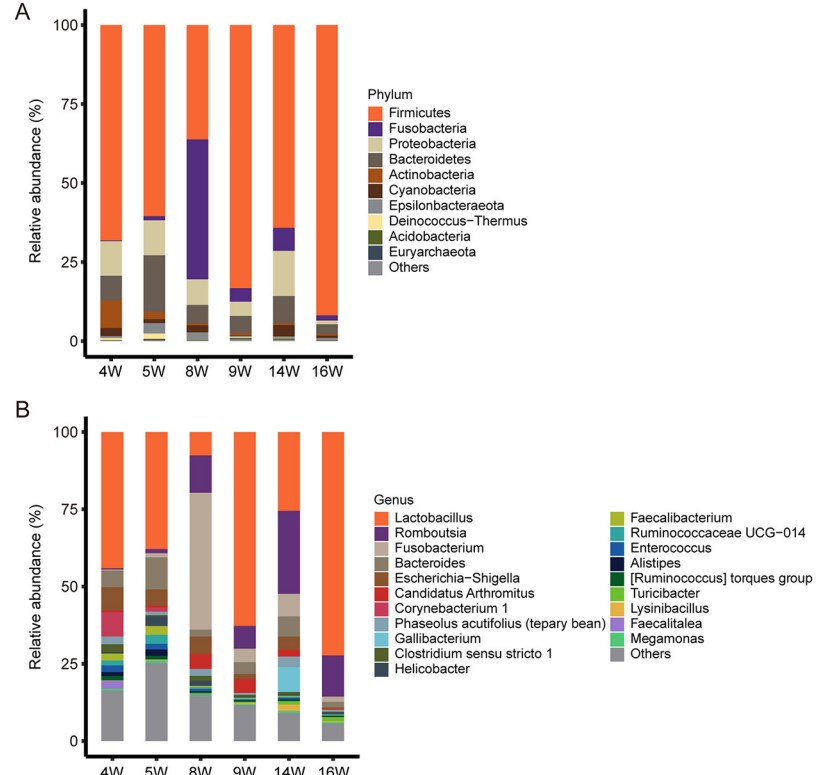

**FIG 2** The microbial composition at the phylum level and genus level. Each bar represents the average relative abundance of each taxon within a time point. (A) The top 10 microbes in relative abundance in feces at the phylum level; the phyla with lower relative abundance are classified as "Others." (B) The top 20 microbes in relative abundance in feces at the genus level; the genera with lower relative abundance are classified as "Others."

categorized as high body weight (HW) ($n = 5$) and the five samples with the lowest body weight in week 9 were categorized as low body weight (LW) ($n = 5$). The weight difference between HW and LW was significant (Fig. S6A and B), indicating that the grouping was reliable. Although not significant ($P > 0.05$), PCoA based on weighted Bray-Curtis distances showed a trend toward separation (Fig. 4A). Then, we used Faith's PD and Shannon indices to visualize the microbiota diversity in two groups. High-weight broilers had a lower diversity, but the difference was not significant (Mann-Whitney, $P > 0.05$) (Fig. 4B and C).

Next, we explored changes in the microbial composition of the two groups. At the phylum level, *Firmicutes*, *Bacteroidetes*, *Proteobacteria*, and *Fusobacteria* were dominant phyla in both HW and LW (relative abundance of >2%) (Fig. 4D; Table S6). At the genus level, *Lactobacillus* was the most abundant genus in two groups, with relative abundances of 62.9% from HW and 62.62% from LW. Besides, *Romboutsia* (HW, 5.51%; LW, 9.27%), "*Candidatus* Arthromitus" (HW, 7.75%; LW, 1.19%), *Fusobacterium* (HW, 5.58%; LW, 3.02%), *Bacteroides* (HW, 3.36%; LW, 4.37%), and *Escherichia-Shigella* (HW, 1.85%; LW, 1.3%) also had a high-abundance occupancy in HW and LW (Fig. 4E; Table S6).

**Microbes associated with body weight.** To explore the microbes associated with broiler body weight, we calculated the Spearman correlation coefficients between the relative abundances of the top 100 (from all 60 samples) genera and individual body weight. Three genera with significant positive correlation with individual weight were selected ($P < 0.01$, FDR < 0.01, and $r > 0.6$), and 13 genera with significant negative correlation with individual weight were selected ($P < 0.01$, FDR < 0.01, and $r < -0.6$) (Table S7). These genera significantly associated with body weight were concentrated in *Firmicutes*, *Actinobacteria*, *Proteobacteria*, *Bacteroidetes*, and *Euryarchaeota*. Among these body weight-related genera, we further screened out 4 genera with high relative abundance (relative abundance of >1%), and they were *Romboutsia* (10.24%), *Corynebacterium 1* (1.66%), *Gallibacterium* (1.42%), and *Faecalibacterium* (1.04%).

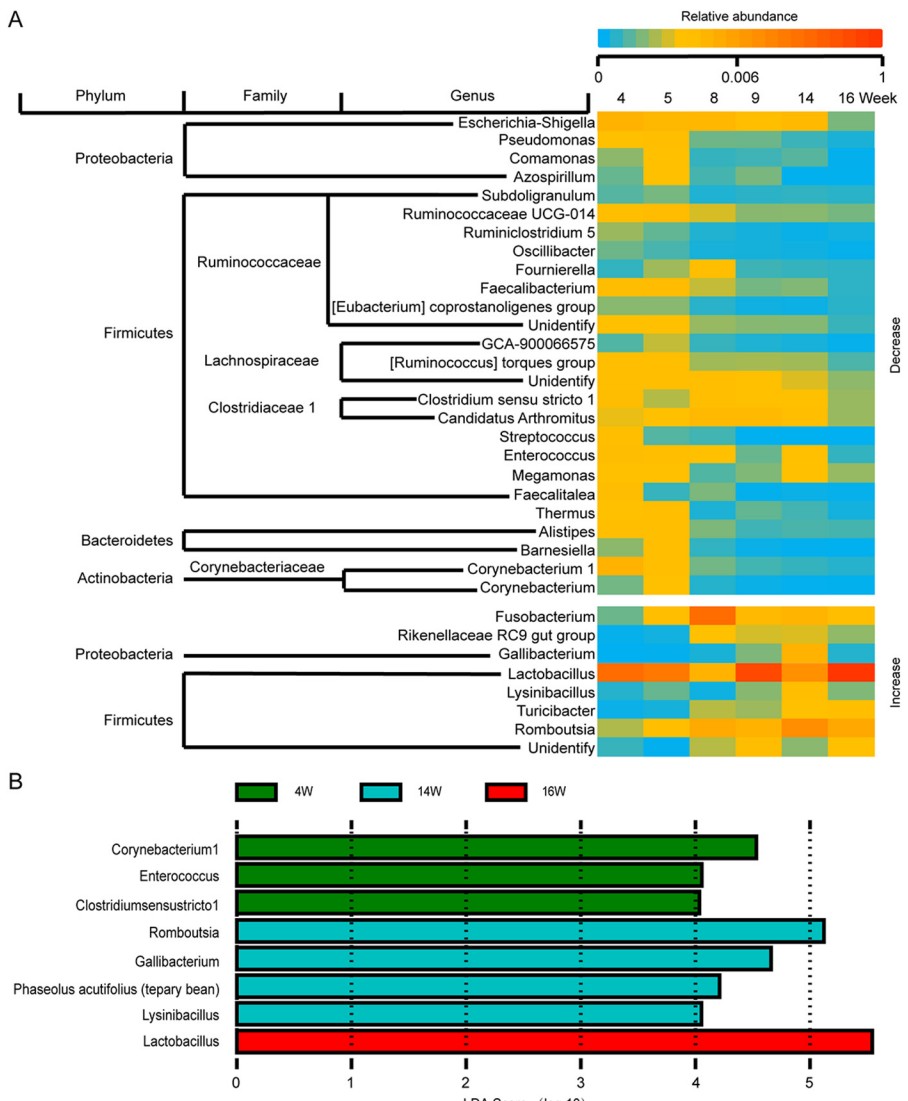

**FIG 3** Changes of flora abundance and microbial biomarkers at different time points. (A) The genera whose average relative abundance within each time point was significantly correlated with different sampling time points (4, 5, 8, 9, 14, and 16 weeks) (Pearson, $P < 0.05$ and FDR $< 0.25$). (B) Eight microbial biomarkers from three groups (LDA $> 4$ and $P < 0.05$). Different colors depict different time groups.

LEfSe analysis between HW and LW based on the relative abundance of all genera revealed potential effects of *Bilophila*, *Epulopiscium*, *Akkermansia*, "*Candidatus* Arthromitus," *Ruminococcaceae UCG-013*, and *Turicibacter* on the body weight of broilers ($P < 0.05$ and LDA score of $>3$) (Fig. 5).

**Predicted function of the broiler gut microbiota.** To explain how intestinal microorganisms affect the growth and development of the host, we next used PICRUSt2 to explore the functional profiles of the chicken gut microbiota. The 16S feature metadata of all samples were loaded for metagenome prediction, and predicted genes were studied for functional annotation and pathway attribution analysis based on KEGG databases. We acquired a total of 7,647 Kyoto Encyclopedia of Genes and Genomes (KEGG) orthologous groups (KOs) and identified 298 pathways in 60 fecal samples. These pathways can explain how gut microbiota affects the growth of broilers to a certain extent.

A constrained principal-coordinate analysis (CPCoA) plot based on Bray-Curtis distances constrained by sampling time points showed that the function of the microbiota in the gut also changed significantly in different time groups (Fig. S2). Microbe-related pathways had the highest enrichment in the metabolism class. The pathway annotation revealed that

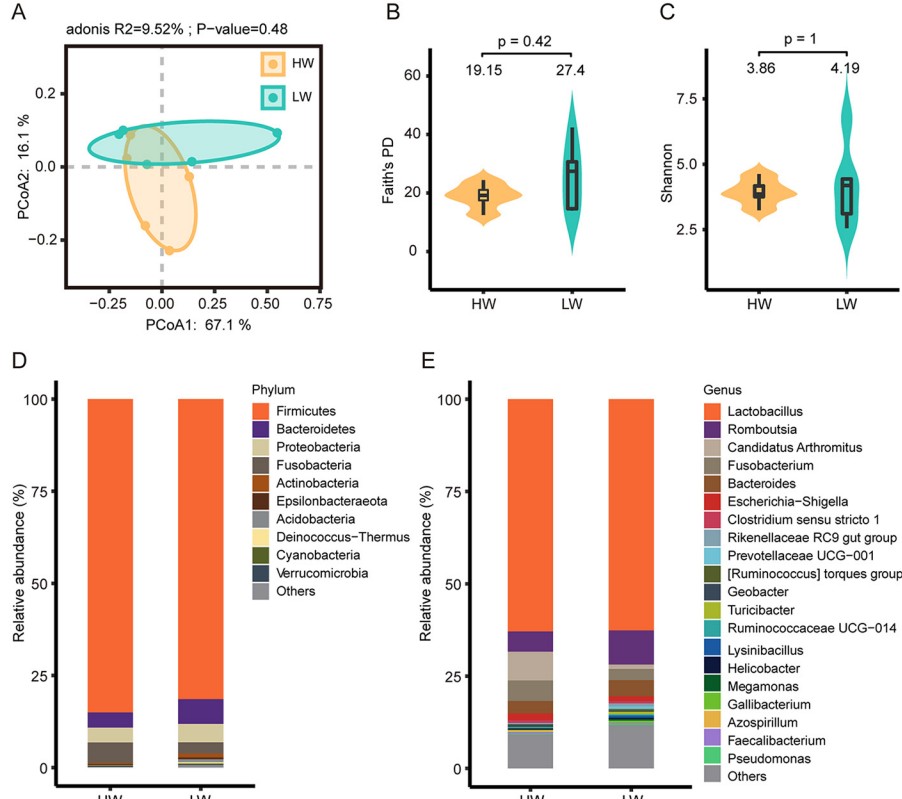

**FIG 4** Microbial diversity and composition in HW and LW. (A) Principal-coordinate analysis (PCoA) based on weighted Bray-Curtis distances. PERMANOVA was used to detect the significance of separation between groups. (B and C) Faith's PD and Shannon index in HW and LW. The box plot shows the quartile, median, and extreme limit of the value. The violin plot shows the distribution of values. The numbers above the violin plots represent the median values. The Mann-Whitney test was used to test the significance of differences. (D) The top 10 microbes with the highest relative abundances in HW and LW at the phylum level; the phyla with lower relative abundance are classified as "Others." (E) The top 20 microbes with the highest relative abundances in HW and LW at the genus level; the genera with lower relative abundance are classified as "Others."

49.13% of the total identified 16S rRNA genes were enriched in pathways related to metabolism, followed by pathways related to genetic information processing, environmental information processing, and unclassified, accounting for 22.81%, 15.12%, and 6.39% of the total enriched genes, respectively. In addition, membrane transport, carbohydrate metabolism, replication and repair, amino acid metabolism, energy metabolism, and translation were identified as the main functional pathways in the secondary classification, accounting for 13.20%, 10.96%, 9.19%, 8.60%, 6.24%, and 5.89% of genes, respectively (Table S8). In addition, we observed that almost all of the top 20 prediction pathways were enhanced in weeks 9 and 16 (Fig. 6A).

Furthermore, 14 KEGG pathways were significantly associated with individual weight among the top 100 (from all 60 samples) level 3 pathways ($P < 0.01$ and FDR of $<0.1$);

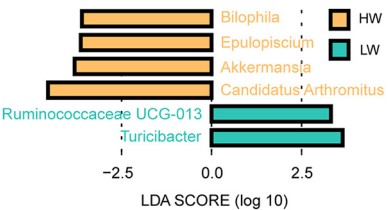

**FIG 5** Genera with significant difference between HW and LW identified by LEfSe analysis in week 9 (LDA > 3 and $P < 0.05$).

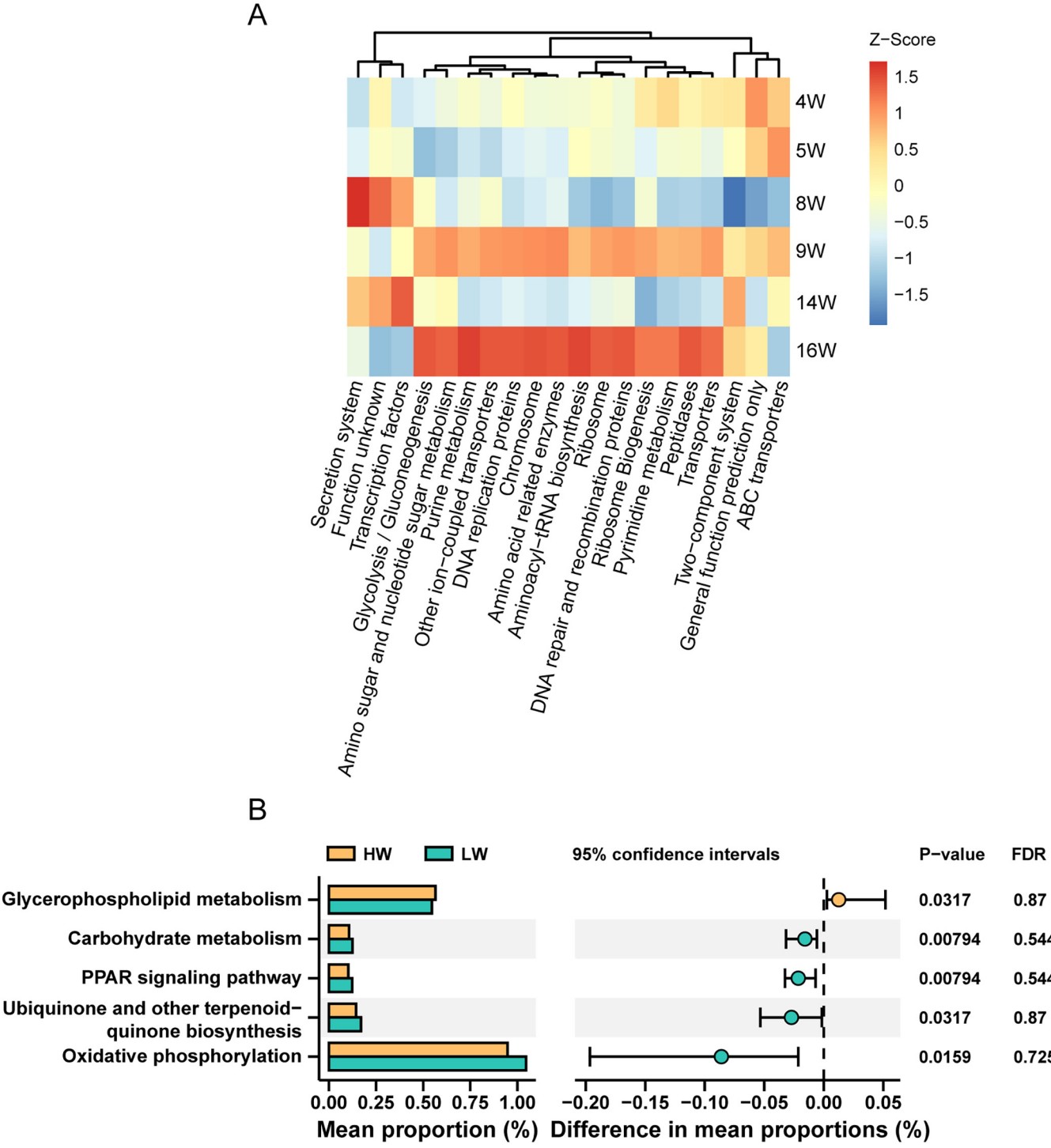

**FIG 6** Pathway analysis of microbiota. (A) Z-score heatmap of the top 20 (from all samples) enriched pathways obtained in the KEGG pathway (Z-score by column). (B) The significantly differential pathways between HW and LW (Mann-Whitney, $P < 0.05$ but FDR > 0.25).

they were mainly distributed in a variety of metabolic pathways and some information processing pathways (Table S9). Through the Mann-Whitney test conducted with STAMP software in week 9, we found 5 differential pathways between HW and LW ($P < 0.05$ but FDR of >0.25) (Fig. 6B). Glycerophospholipid metabolism was significantly upregulated in HW. Carbohydrate metabolism, peroxisome proliferator-activated receptor (PPAR) signaling pathway, ubiquinone and other terpenoid-quinone biosynthesis, and oxidative phosphorylation were significantly upregulated in LW. In total, these 19 pathways (Fig. 6B; Table S9) indicate

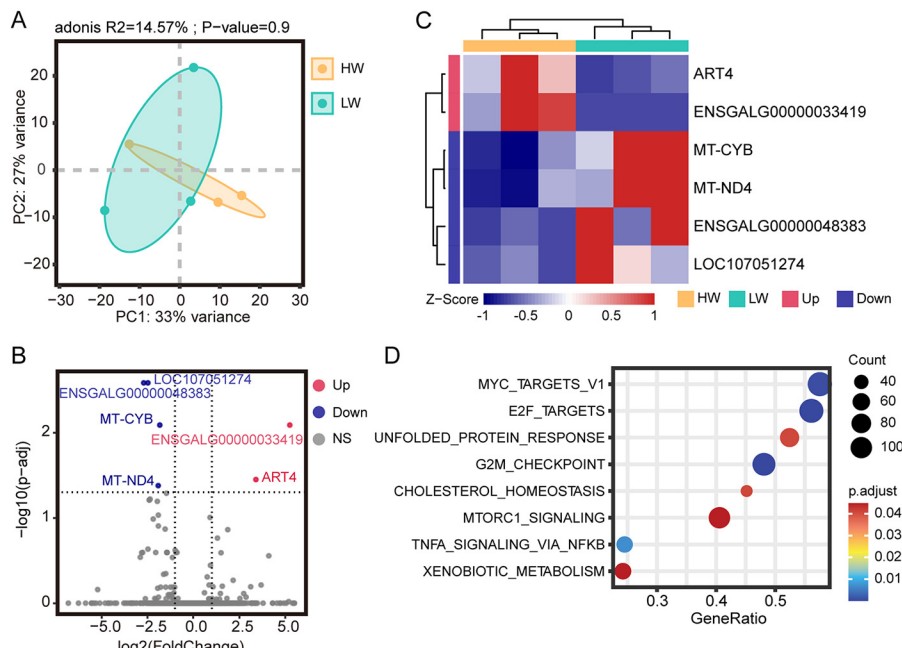

**FIG 7** Transcriptomic analysis of differentially expressed genes in cecum between HW and LW. (A) Principal-component analysis (PCA) based on gene expression as transcripts per million (TPM). The dashed range represents the smallest ellipse surrounding all samples in each group. PERMANOVA was used to detect the significance of separation between groups. (B) Volcano plot of differentially expressed genes between the two groups. (C) Z-score heatmap of differentially expressed genes (Z-score by row). Each column represents a sample, and each row represents a gene. (D) Bubble plot of GSEA enrichment results (FDR, $P_{adj} < 0.05$).

that metabolic and information processing-related pathways may be potential pathways for intestinal flora to affect broiler weight.

**Transcriptome analysis of cecal tissue in high- and low-weight broilers.** To verify the effect of broiler gut microbiota on the body weight of broilers, we divided 6 randomly selected broilers at week 9 into a high-weight group (HW) ($n = 3$) and a low-weight group (LW) ($n = 3$). The weight difference between HW and LW was close to significant ($P = 0.07$) (Fig. S6C and D), indicating that the grouping was reliable. We then performed transcriptome sequencing analysis of the broilers' cecal tissues. After filtering out reads containing adaptor sequences and reads with low quality, a total of 40.18 Gb of high-quality reads was obtained, corresponding to an average of ~6.7 Gb per sample. The high-quality reads were mapped to the chicken reference genome (Gallus-6.0) with a mapping rate of 94.14 to 95.24% (Table S8). Based on the gene expression dynamics, the Spearman correlation coefficients between samples were calculated (Fig. S3). In addition, the results of principal-component analysis (PCA) showed that there was no significant difference (PERMANOVA, $P > 0.05$) in the overall expression level between HW and LW (Fig. 7A).

To further investigate the transcriptional differences of cecal tissue between HW and LW, differential expression analysis with LW as control was performed. Six significantly differentially expressed genes (DEGs) were screened using $P_{adj}$ of <0.05 and |log$_2$ (fold change)| of >1 as the standards, including 2 upregulated genes and 4 downregulated genes (Fig. 7B and C). Upregulated genes include *ART4* and *ENSGALG00000033419*, and downregulated genes include *MT-CYB*, *MT-ND4*, *ENSGALG00000048383*, and *LOC107051274*. These genes are mainly related to energy metabolism (*ART4*, *MT-CYB*, and *MT-ND4*) and immunity (*LOC107051274* and *ENSGALG00000048383*). Also, "Immunoglobulin production" is the only Gene Ontology (GO) term enriched with DEGs (Fig. S4). Through gene set enrichment analysis (GSEA), 8 pathways were found to be enriched in two BW groups (Fig. 7D). Seven pathways (such as "unfolded protein response," "G$_2$M checkpoint," and "mTORC1 signaling") were enriched in LW (Fig. S5A to G), and "xenobiotic metabolism" was enriched in HW (Fig. S5H). These results suggested that there may be differences in immune and energy metabolism levels between HW and LW and that LW may have a higher level of cellular metabolism.

## DISCUSSION

The gut microbes of poultry are a large and dynamic community. Some previous studies focused on the dynamic changes in gut microbes in the early stages of chicken development and stated that the diversity gradually increased until approximately 4 weeks of age (23, 24). However, high-quality local broiler breeds usually have a long feeding cycle. Therefore, this research on dynamic changes in broiler gut microbes after 4 weeks is of significance. In terms of intestinal microbial diversity, in humans, alpha diversity gradually increases with growth and development before the age of 5 years; however, there is a decrease in gut microbial diversity from 18 to 80 years of age (26, 27). Some investigators studied the intestinal flora of pigs on days 1, 28, and 70 and found that with the increase of age, the microbial diversity of the ileum increased, the proportion of *Firmicutes* increased, and the proportion of *Proteobacteria* decreased in Jinfen white pigs and Mashen pigs (28). Microbial diversity tends to increase in chicken embryos up to week 4, peaking at approximately days 14 and 28 for the foregut and hindgut, respectively, and then remaining stable or decreasing slightly thereafter (22, 23). In agreement with previous studies, we found that broiler gut microbial alpha diversity showed a decreasing trend from weeks 4 to 16. This is a supplement to previous research that found, overall, increased relative abundances of *Firmicutes* and decreased relative abundances of *Proteobacteria*. These data indicate large changes in the gut microbiota and its composition in broilers after 4 weeks of age and appear to portend a common changing trend of gut microbiota across multiple species as well as shared dominant phyla.

Animals and plants are holobionts composed of the host and many symbiotic microorganisms (29). Some researchers believe that the holobiont with its hologenome acts as a unique biological entity and therefore also as a level of selection in evolution (30, 31). The enhancement of almost all of the top 20 predicted pathways at weeks 9 and 16 may reflect the effect of artificial selection on the holobiont, as these functions were significantly enhanced 1 week prior to slaughter day. When microbial competition inhibits the cooperative network and increases stability, the host can benefit from it, accompanied by a decrease in flora diversity (32). The improvement of the immune system also affects the diversity of microbial species, especially in the suppression of pathogenic bacteria (29). In this study, gut microbial diversity decreased with broiler growth, perhaps due to flora competition and host immune modulation. The relative abundance of *Fusobacteria* at the phylum level and *Fusobacterium* at the genus level sharply increased in week 8 (44.25%) and returned to baseline in week 9 (4.3%), which suggested the competition and self-regulation of gut flora.

At present, the composition of chicken intestinal microbiota has been proved to be closely related to host production and health (33–35). Some studies have been carried out to investigate the relationship between intestinal microbiota and body weight in broilers (36–38). Increased microbial competition leads to reduced diversity and more benefits for the host (32). We found that HW had a lower alpha diversity, although not significant ($P > 0.05$), which is consistent with findings in 300-day-old laying hens (39). At the phylum level, obese animals and humans exhibit an increased abundance of *Firmicutes* and a decreased abundance of *Bacteroidetes* (40). In this study, the ratio of *Firmicutes* to *Bacteroidetes* in HW was higher than that in LW, which was consistent with previous studies (39, 40), indicating that these may be common phenomena.

*Bilophila* was associated with obesity in mice induced by a high-fat diet (41), but it was also found to be significantly negatively correlated with broiler weight (42). As a potential beneficial bacterium, *Akkermansia* was more abundant in the better-fed efficacy group hens (43). *Akkermansia muciniphila* can prevent body weight loss by alleviating intestinal mucosal damage caused by dysentery in chickens (44). And dietary supplementation with sophorolipid could significantly increase the feed utilization and increase the abundance of *Akkermansia muciniphila* in chicks (45). The abundance of "*Candidatus* Arthromitus" decreased during the period of broiler infection with necrotic enteritis (46), indicating that "*Candidatus* Arthromitus" may be a beneficial bacterium for broilers. In this study, we found that *Bilophila*, *Epulopiscium*, *Akkermansia*, and "*Candidatus* Arthromitus" were more abundant in HW. Combined with the results of us and other researchers cited above, we speculated that

*Akkermansia* and "*Candidatus* Arthromitus" are conducive to the healthy weight gain of broilers, and *Bilophila* needs to be further studied in regard to the relationship with the weight gain of broilers.

Through Spearman correlation analysis, we screened 16 genera and 14 pathways that might have an effect on broiler body weight. Among the 16 genera, *Romboutsia*, *Gallibacterium*, and *Methanobrevibacter* were significantly positively correlated with body weight. *Methanobrevibacter* was found to promote abdominal fat deposition in broilers (38). Dietary supplementation with *Bacillus subtilis* improved growth performance of broilers, which may be related to the increased abundance of *Romboutsia* (47). However, the severe respiratory infection caused by *Gallibacterium anatis* will seriously affect the laying by hens (48). Combined with the results listed above, we should conduct more in-depth studies on the mode of action of genera significantly associated with body weight. At the same time, it should be noted that the genera significantly related to weight identified by Spearman correlation in this study may have an indirect impact on weight, due to both weight and bacterial composition being associated with age.

In this study, the PPAR signaling pathway, ubiquinone and other terpenoid-quinone biosynthesis, carbohydrate metabolism, and other microbial functional pathways that differed between HW and LW were basically related to microbial carbohydrate metabolism and lipid metabolism. Transcriptome analysis showed that the differentially expressed genes between HW and LW were related to immunity (*LOC107051274* and *ENSGALG00000048383*) and energy metabolism (*ART4*, *MT-CYB*, and *MT-ND4*). Therefore, we speculated that the intestinal microorganisms of broilers may affect the energy metabolism of the host cecum through their own carbohydrate metabolism and lipid metabolism.

In summary, our study showed that the intestinal microbial diversity and composition of broilers still changed dynamically after the 4th week, and the microbial alpha diversity showed a downward trend. Although there was no significant difference in intestinal microbial alpha diversity between high-weight and low-weight broilers, there were changes in microbial composition. In addition, we found a significant positive correlation between the relative abundance of some genera and the body weight of broilers or significant differences between high- and low-body-weight broilers. Although the mechanism of influence needs further research, combined with the functional analysis of microorganisms and the transcriptome results of cecal tissue, we speculated that microorganisms may affect the immune level and energy metabolism level of broilers through their own carbohydrate metabolism and lipid metabolism and then affect body weight.

## MATERIALS AND METHODS

**Animals and experimental design.** The broilers used in this experiment included 120 male broilers (Tianfu broilers) hatched and raised on the poultry farm of Sichuan Agricultural University (Ya'an, China). Broilers were subjected to the same feeding regimen and environment, were housed separately in cages, and had free access to commercial broiler fodder and water. The room temperature was maintained at 32°C during the 1st week of the experiment and then gradually decreased to ambient temperature until the end of the experiment. Starting from the 4th week, the body weight of all individuals was measured every week. Ten individuals from 120 broilers in weeks 4, 5, 8, 9, 14, and 16 were randomly selected for collection of fecal samples by swab and not returned after sampling (49). Cecal tissue samples were taken from 6 broilers randomly selected in week 9. All samples were stored at −80°C until further analysis.

**DNA extraction and microbial 16S rRNA gene sequencing analysis.** Fecal microbial DNA was extracted using the Ezup oral swab genomic DNA extraction kit (Sangon Biotech, Shanghai, China) following the manufacturer's guidelines. The extracted DNA was quantified using a NanoDrop2000 spectrophotometer (Thermo Fisher Scientific, DE, USA), and the quality was assessed by 1% agarose gel electrophoresis. The V3-V4 hypervariable regions of the bacterial 16S rRNA gene were amplified using specific primers 341F (5′-CCTAYGGGRBGCASCAG-3′) and 806R (5′-GGACTACNNGGGTATCTAAT-3′) (50, 51). All PCRs were carried out with 15 $\mu$L of Phusion high-fidelity PCR master mix (New England Biolabs), 0.2 $\mu$M forward and reverse primers, and approximately 10 ng template DNA. Thermal cycling consisted of the initial denaturation at 98°C for 1 min, followed by 30 cycles of denaturation at 98°C for 10 s, annealing at 50°C for 30 s, and elongation at 72°C for 30 s, and then, finally 72°C for 5 min. PCR products were purified with a Qiagen gel extraction kit (Qiagen, Germany). Sequencing libraries were generated using the TruSeq DNA PCR-free sample preparation kit (Illumina, USA) following the manufacturer's recommendations, and index codes were added (see Table S13 in the supplemental material for primer sequence and barcode sequence). The library quality was assessed on the Qubit 2.0 fluorometer (Thermo Scientific) and Agilent Bioanalyzer 2100 system. Finally, the library was sequenced on an Illumina NovaSeq platform, and 250-bp paired-end reads were generated.

The 16S rRNA gene sequencing data analysis was performed using Quantitative Insights Into Microbial Ecology 2 (QIIME2) software (52). The DADA2 method was used for sequence quality control. Barcode and primer sequences were removed from the 5′ ends, and low-quality bases were truncated from the 3′ ends according to the Q20 and Q30 standards. The clean FASTA sequences overlapped, and then the feature table was constructed after deredundancy. The final high-quality representative feature sequences were used for taxonomic annotation with the SILVA database (132_99 release) (53, 54).

For metrics of taxonomic diversity, we used Shannon and Faith's phylogenetic diversity (Faith's PD) (55) indices to compute the alpha diversity, and different time groups were compared using the Kruskal-Wallis H test. The comparisons of gut microbiota compositions between time groups in the growth process were conducted by univariate permutational analysis of variance (PERMANOVA) for the Bray-Curtis distances and illustrated by principal-coordinate analysis (PCoA).

We referred to the SILVA database to obtain the taxonomic composition of microorganisms in the QIIME2 analysis process, and default parameters (qiime feature-classifier classify-sklearn --i-reads --i-classifier --o-classification) were used. Linear discriminant analysis (LDA) coupled with linear discriminant analysis effect size (LEfSe) was used to identify differentially abundant taxa among various time groups and between HW and LW at the genus level (56). The Spearman correlation coefficient was calculated to measure the correlation between individual microbial abundance and body weight and between KEGG pathway and body weight. The Pearson correlation coefficient between the average relative abundance of each genus in all groups and the sampling time points was calculated. $P$ of <0.05 and FDR of <0.25 were considered to be the thresholds of significant correlation.

Phylogenetic investigation of communities by reconstruction of unobserved states (PICRUSt2) was used to infer the functional potential of fecal microbiota and further compared the functional profiles at various time points by Statistical Analysis of Metagenomic Profiles (STAMP) software (57, 58). On the ImageGP platform (59), weighted Bray-Curtis distances were calculated based on the relative abundance of PICRUSt2-predicted KEGG pathways, and constrained principal coordinate analysis (CPCoA) of all samples was performed.

**RNA extraction and transcriptome analysis.** Total RNA was extracted from cecum tissue using RNAiso Plus total RNA extraction reagent (TaKaRa) following the manufacturer's instructions, and RNA was detected with a NanoDrop2000 spectrophotometer (Thermo Fisher Scientific, DE, USA). The quantity of total RNA was determined using a Qubit 2.0 fluorometer (Life Technologies, CA, United States), and the integrity of total RNA was analyzed using a Bioanalyzer 2100 system (Agilent Technologies, CA, USA). After the RNA samples were qualified, the eukaryotic mRNA was enriched by magnetic beads with oligo(dT). Then, cDNA was synthesized using mRNA as the template. Second-strand cDNA was synthesized by buffer, deoxynucleoside triphosphates (dNTPs), DNA polymerase I, and RNase H and purified using AMPure XP beads. The purified double-stranded cDNA was end repaired; poly(A) was added, and the cDNA was ligated to a sequencing connector. After fragment selection and PCR amplification, a sequencing library was obtained and sequenced using DNBSEQ-T7 by Novogene (Beijing, China).

We removed low-quality reads and calculated the Q20, Q30, and GC contents of the clean reads to obtain high-quality clean reads. Clean data were mapped to the chicken reference genome (Gallus-6.0) using STAR (v2.7.6a) (60). Then, Kallisto (v0.44.0) (61) software was used to quantify gene expression as transcripts per million (TPM). Benjamini and Hochberg's procedure was used to adjust the value of $P$ (62, 63). The differentially expressed genes (DEGs) with an adjusted value of $P$ ($P_{adj}$) of <0.05 and $|\log_2 (\text{fold change})|$ of >1 identified by DESeq2 (v1.30.1) (64) were considered differentially expressed. Functional enrichment including GO terms and KEGG pathways of differentially expressed genes was performed by Metascape (65), and $P$ of <0.05 was considered significant. Rank-based gene set enrichment analysis (GSEA) was performed using the clusterProfiler (v4.0.5) (66) package and hallmark gene sets from MSigDB (http://www.gsea-msigdb.org/gsea/msigdb/), and the significant enrichment pathways with $P_{adj}$ (FDR) of <0.05 and $|$normalized enrichment score (NES)$|$ of >1 were identified.

**Ethical approval.** The complete procedure in this study was approved by the Committee on the Care and Use of Laboratory Animals of the State-Level Animal Experimental Teaching Demonstration Center of Sichuan Agricultural University.

**Data availability.** Original data sets are available in a publicly accessible repository: The original contributions presented in the study are publicly available. These data can be found at NCBI SRA (https://www.ncbi.nlm.nih.gov/sra) and under BioProject identifier (ID) PRJNA779029.

## SUPPLEMENTAL MATERIAL

Supplemental material is available online only.
**SUPPLEMENTAL FILE 1**, PDF file, 0.5 MB.
**SUPPLEMENTAL FILE 2**, XLSX file, 0.3 MB.

## ACKNOWLEDGMENTS

This work was supported by a grant from the Sichuan Science and Technology Program (2019JDTD0009).

We declare no conflict of interest.

D. Li and H. Jie contributed to conception and design of the study and provided critical suggestions and comments. L. Shi and Y. Ge performed the experiments. M. Yang and L. Shi performed the bioinformatic analyses. M. Yang, L. Shi, and D. Leng wrote and revised the manuscript. All authors reviewed and approved the final manuscript.

Thanks to all members of the laboratory for their valuable comments on the manuscript. We also appreciate the High-Performance Computing Platform of Sichuan Agricultural University for providing the powerful support for data analysis.

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
