## [Reviewer comments · Microbiology Spectrum]

Microbiology Spectrum

Dynamic changes in the gut microbial community and function during broiler growth

Maosen Yang, Lianzhe Shi, Yile Ge, Tao Wang, Hang Jie, Diyan Li, Dong Leng, and Bo Zeng

Corresponding Author(s): Diyan Li, Sichuan Agricultural University

Review Timeline:

Submission Date:	March 19, 2022
Editorial Decision:	June 8, 2022
Revision Received:	July 23, 2022
Accepted:	July 26, 2022

Editor: John Atack

Reviewer(s): Disclosure of reviewer identity is with reference to reviewer comments included in decision letter(s). The following individuals involved in review of your submission have agreed to reveal their identity: Stanley H. Chen (Reviewer #2); Rabindra Kumar Mandal (Reviewer #3); Gajender Aleti (Reviewer #4)

Transaction Report:

DOI: <https://doi.org/10.1128/spectrum.01005-22>

June 8, 2022

Prof. Diyan Li
Sichuan Agricultural University
College of Animal Science and Technology
Chengdu
China

Re: Spectrum01005-22 (Dynamic changes in the gut microbial community and function during broiler growth)

Dear Prof. Diyan Li:

Link Not Available

Sincerely,

John Atack

Journals Department
Reviewer comments:

Reviewer #1 (Comments for the Author):

The authors examined the fecal microbiome of broiler chickens on multiple timepoints (4-16 weeks) using 16S amplicon sequencing. They show a difference in the microbial composition during the different timepoints, including a decrease in alpha diversity, beta-diversity based clustering and specific bacteria that change. They also try to see the association between body weight and the microbial composition, albeit less convincingly (see specific comments below). They also look at the predicted gene content using picrust2, and again look for age/weight related pathways. Finally, they performed transcriptomics in 6 chickens from a single timepoint, and try to identify host genes associated with body weight. The cohort is large (10 chickens * 6 time points), focuses on a less-studied population (age 4-16 weeks). and results seem clean of major artifacts. However, some of the statistical analysis is less well supported and/or well described (i.e. application of FDR to some analysis, use of t-test,

some observations do not have any statistical support - see specific comments below), and some results are overstated/interpreted.

Specific comments:

Methods:

-
1. What chickens were sampled at each time point? If I understand correctly out of 120 chickens, 10 were randomly chosen at each time point? If so, are there chickens that participated in more than 1 timepoint? What is the reason for this design? Please clarify in the methods section.
 2. were the 16S PCR primers barcoded? how long were the barcodes? there is "and index codes were added." is this the stage where the barcodes were added? If so, how were they added?
 3. was each chicken separately housed? please add to methods section.
 4. "We referred to the SILVA database to obtain the taxonomic composition of microorganisms." - how exactly was taxonomy assigned? using what software/parameters?
 5. "Functional enrichment including GO terms and KEGG pathways of differentially expressed genes was performed by Metascape (88), and $p < 0.05$ was considered as significant." - I think this relates to the 16S picrust analysis? if so, please move to the relevant section in the methods.

Results:

-
6. "and increased from the initial average weight of 304.2 g in week 4 to 3, 105.8 g in week 16" is it supposed to be 3105.8? If so, please write without the space and the comma between "3" and "105.8"
 7. "These results showed that there were significant and dynamic changes in the gut microbial community of broilers after 4 weeks of age." - this sentence is not clear (what is "dynamic changes"?). Would recommend rephrasing or removing
 8. "we correlated the average relative abundance of each time group to feeding time" - what does this mean? what is the "feeding time"? did you mean you identified bacteria correlated with the sample timepoint? how was correlation performed (pearson/spearman/other?)? are the significant bacteria shown in figure 3A? what p-value threshold? was FDR correction applied?
 9. "indicating that the gut microbiota changed dramatically in week 14." I think "dramatically" is a bit of an overstatement. Is the week 14 change much bigger than the previous week changes? if not, maybe rephrase to "indicating that the gut microbiota continues to change even in week 14".
 10. "These results also proved that the gut flora changes dynamically during the growth of broilers." - "proved" is a bit strong. can change to something like "These results indicate" or "These results show".
 11. "Analysis of gut microbial diversity and composition in high and low weight broilers" section: How were the chickens divided into the high and low weight groups? What was the weight cutoff? was it the same in all time points? Also, why are there 23+25=48 chickens total in both groups combined and not 60? Please supply more details (maybe in the methods section).
 12. "However, 3 dimensions principal co-ordinates analysis (3D-PCoA) based on weighted Bray-Curtis distances showed a trend towards separation (Fig. 4A)" I did not manage to see a trend in the PCoA. Do you have some statistic (such as permanova) to support your claim that the bacterial community differs between the high and low groups?
 13. "These results illustrated that grouping was relatively reasonable." not clear to me what is the meaning of this sentence. What is a "reasonable grouping"? What supports your claim that this grouping is reasonable? Maybe delete this sentence or explain better.
 14. "Firmicutes and Proteobacteria were relatively overrepresented in HW," - this whole paragraph has a lot of statements without statistical support. Since we are comparing two finite groups the mean frequency of each bacteria will always be different (i.e. not exactly the same). However, in order to claim that a specific bacteria has a different mean in the 2 populations (and not just in the random subset we are looking at), we need to use some statistical test to reject the null hypothesis (of similar mean in the 2 original groups). Please supply some statistical test results (i.e. lfcse or mann-whitney with FDR correction) or alternatively remove the higher/lower/over-represented/under-represented parts of this paragraph.
 15. "Microbes associated with body weight" - was correlation performed on all 60 samples or a subset? If on all 60 samples, how

do you control/separate the effect of aging (i.e. older chickens weigh more) from the effect of specific bacteria (that supposedly lead to increased weight)? If no control is applied, please add a note in the results and discussion sections mentioning that the association may be indirect, due to both weight and bacterial composition being associated with age.

16. "Microbes associated with body weight" - why did you use pearson correlation and not spearman? Spearman searches for linear correlation, which does not have to be the case with bacteria/weight relations, whereas pearson only looks for monotonic relation (which to me sounds more biologically probable). Note that it is totally ok to use pearson if you prefer, and the results are valid. But maybe spearman will provide more significant bacteria? Note that this is just a recommendation - you do not have to change the manuscript if you prefer pearson.

17. "Microbes associated with body weight" - was the association performed at the genus level? or the ASV level? please clarify. If it was performed at the ASV level, please provide the specific ASV sequences in table S6

18. "Microbes associated with body weight" - was FDR control used for the significant weight-associated bacteria? If not, please apply FDR rather than just using p-values (as you test 100 different bacteria, we would expect 5 to have $p < 0.05$ even if the null hypothesis holds for all bacteria). Note that even if no bacteria show significant association after FDR, it is fine to write so and does not reduce the power of the manuscript.

19. "STAMP analysis with differential relative abundances between HW and LW revealed potential effects" - There is no description in the methods section of how STAMP was used for the HW/LW groups analysis (it only states STAMP was used for picrust2 analysis). Please supply details how STAMP was used - what groups were compared, was there control for age, what statistical test was used, was there multiple hypothesis control, what group of features was compared (100 most common or all, genus or ASV level or all phylogenetic levels) etc.

20. "7647 Kos" - need to capitalize to O - "7647 KOs"

21. "Constrained principal component analysis showed that the function of microbiota in the gut also changed significantly in different time groups" - please give details regarding the software implementation and parameters analyzed using constrained-PCA in the methods section.

22. "The pathway annotation revealed that 49.13% of the total enriched genes were enriched in pathways related to metabolism" - how was the enriched genes identified? using STAMP? what groups were compared? what statistical test was used? what parameters? please provide details.

23. "In addition, membrane transport, carbohydrate metabolism, replication and repair, amino acid metabolism, energy metabolism, translation were identified as the main functional pathways in the secondary classification" - what is the secondary classification? what is the difference from the analysis above? do you mean different kegg levels? please provide details for this analysis as well

24. "Furthermore, 5 pathways were significantly associated with individual weight among the top 100 level 3 pathways, they are selenocompound metabolism, chaperones and folding catalysts, streptomycin biosynthesis, thiamine metabolism and carbon fixation pathways in prokaryotes, respectively ($p < 0.05$)" - how were these pathways identified? pearson correlation with weight? please specify. Also, was multiple hypothesis correction (FDR) used (since the test was performed on 100 pathways, we would expect 5 with $p < 0.05$ even if the null hypothesis is correct for all pathways)? If not, please use FDR or another multiple hypothesis control method (ok to use $FDR < 0.1$ or even $FDR < 0.25$).

25. "Through Welch's T test conducted with STAMP software, we found 4 differential pathways between HW and LW " - what groups of chickens did you use for the test? the same 23+25 as used in the 16S analysis? please provide details. Also, did you control for age as an effector of the pathway abundance? (see comment above regarding effect of age on weight)

26. "Through Welch's T test conducted with STAMP software, we found 4 differential pathways between HW and LW ($p < 0.05$)" - i do not think t-test is appropriate here, as it is a parameteric test that assumes normality. Can you show the pathway abundance follows a somewhat normal distribution rather than zero-inflated? Otherwise, please use Mann-Whitney instead. Also, did you use multiple-hypothesis correction (FDR)? If not, please apply FDR, as you are testing multiple pathways (can use a threshold of 0.1 or even 0.25). If nothing turns out significant, it is still ok and does not reduce the importance of your manuscript.

27. "The results showed that the biological duplicate samples in the same group have relatively high correlation coefficients with each other" - what are the "biological duplicates"? Do you mean samples from the same weight group (LW/HW)? Please clarify.

28. "The results showed that the biological duplicate samples in the same group have relatively high correlation coefficients with each other" - I did not see a strong difference between the within-group vs. between group sample correlations (fig S3). Do you have some calculation to show the within group correlation is significantly higher than the between-group correlation? If not, would recommend to remove this sentence.

29. "Furthermore, the Principal components analysis (PCA) results showed that there was a certain degree of variance between HW and LW (Fig. 7A)." - What does "a certain degree of variance between HW and LW" mean? please clarify. Hard for me to see a difference between the 2 groups based on Fig7A. Do you have some statistic (for example permanova) to show that there is a separation between the HW and LW groups? otherwise, please delete this sentence and the following sentence (" These results indicated that there were differences in gene expression in cecum of broilers with different body weights.")

Discussion:

30. "In this study, gut microbial diversity decreased and stabilized with broilers growth" - What supports the claim that the diversity "stabilized" with broilers growth? Please clarify or remove the "stabilized" part

31. "which illustrated to some extent that the gut microbiota may be affected by multiple factors which from real environment and embodied the cooperation and competition of flora". I did not understand this sentence.

32. "We found that HW had higher alpha diversity than LW, which is different from the study in laying hens" - as this result was not significant (or even close to significant), it could be due to chance as the null hypothesis is not rejected with high confidence. Please write that the results was not significant, or remove.

33. "Minerals is considered micronutrients" - correct to "Minerals are considered micronutrients"

34. "indicating that microorganisms in high-weight broilers consumed more dietary selenium" - i think this is a bit unsubstantiated. Since picrust2 predictions are at the gene level, it only indicates that there are more bacteria with the potential for selenium metabolism. The hypothesis that high-weight broilers consumed more dietary selenium is a possible, but not the only explanation.

Figures:

35. fig 1B: was rarefaction applied before calculating Faiths PD? if so, to what depth? If not, please rarify prior to calculating Faith PD, as it is a binary metric, which may be affected by read depth.

36. fig 2B: I am a bit confused as the legend contains many (34) colors, but the bars in the plot seem to contain a much smaller number of colors?

37. fig 3: "with feeding time" - what is "feeding time" - do you mean the sample week (4/5/8/9/14/16)? please clarify.

38. fig 5: please supply more details in the legend (i.e. what software/statistical test was used, what groups were compared etc.)

39. fig 6B: please supply more details in the legend - how was significance calculated? what test/parameters?

40. fig S6: please specify what each sampleid means (i'm assuming 1_X is the LW group, 2_X is the HW)? Please clarify in the legend if so (or another explanation)

Data availability:

41. The accession provided in the text is good. If possible, please add the per-chicken weight to the metadata, and specify also the per-sample timepoint (i.e. age weeks) in addition to collection date.

Reviewer #2 (Public repository details (Required)):

The 16S microbiota dataset needs to be uploaded to the public repository.

Reviewer #2 (Comments for the Author):

The authors presented a study on the broiler gut microbiota in terms of its changes over the rearing period from 4 weeks to 16 weeks of age. The study offers insights into the composition of gut microbiota between high and low weight broilers and predicts the functions related to the weight gain/loss. It could provide useful information into the broiler production.

Despite some minor grammar issues, the manuscript is reasonably well written. It is suggested to shorten some content in the discussion section. The discussion seems to be a bit of wordy.

L44: why "has" is crossed over?

L45: Read the journal guideline about how to do in-text citation. Similar issues noted throughout the manuscript, please correct them accordingly.

L51: what management are you referring here? Animal welfare, feed conversion ratio, etc.?

L53: His -> Their. It's "Fredrik et al.", not "Fredrik".

L77: As the way you present is the Results section first then the Methods, I suggest to provide a brief summary of what the samples were, the time points, etc. so it's easier to understand what you're assessing.

L89: You should have the full term spelled out when they are referred to for the first time, e.g. PCoA.

L91: Suggest changing the expression to "except samples of Week 4 and 5, and Week 9 and 16", or "except between Week 4 vs. Week 5, and Week 9 vs. Week 16", to avoid the confusion.

L96: It is important to provide the version number of SILVA that you accessed for 16S analysis.

L97: Top 10 from each time point or all together? Same with Top 33.

L102: Due to the colour similarity of the stacked bar chart making it hard to tell the accurate proportion of taxa, I would suggest providing the detail of the relative abundances when certain organisms are listed at various time points. For instance, the proportions of *Lactobacillus* over some weeks collection are hard to compare. Same with other taxa you have listed here. We don't know to what extend the high level or low level was in the dataset.

L108: Similar to the previous comment, details are needed to examine the change of microbiota over time. Just stating significant with a p value nor the exact proportion isn't enough for assessment. Same applies to the following genera changes in Week 14. What were they? How much were they changed at which time points?

On a personal note, it is a problem publishing microbiome papers when detailed information is missing from all the colourful charts in the results section. The data visualisation is fancy and colourful, but it doesn't provide the robust assessment for the purpose of scientific evaluation. People won't be able to tell the exact abundance of taxa from all the colourful bars, let alone if some researchers are colour blind, which isn't a rare case.

L121: Honestly it's hard to tell the separation from the 3D illustration. Unless you can rotate it to find a better viewing point to show the separation, it's not a good way to present data in most cases. A PCoA with clustering is a better option.

L127: Again, how much did they present in LW? You should provide a good comparison between HW and LW in terms of some phyla if you want to state their different abundance in two sample groups. Listing them in HW without giving their abundance in LW doesn't help with the evaluation.

L130: From the taxa below, I assume 43.34% is from HW. But it would be better to have it more specific and clearer.

L140: What were they averaged abundance over time?

L145: It is important to highlight their roles in terms of positive or negative impact on the weight gain/loss.

L165: were

L173: Although the results of transcriptome of broiler cecum yielded some interesting indications, the sample size and the result isn't sufficient or significant to draw any meaningful conclusion. Focus on the microbiota to avoid distraction of the study purpose.

L210: And -> and, same in L213. Minor grammar mistakes.

L215: If choosing to remove the transcriptome analysis, this paragraph should be altered. Focus on the microbiota of your data and its possible implication.

L228-231: This is a very long sentence, with too many "which". Break it up into a few sentences with a logic flow is better.

L237-238: This speculation is a bit of too stretched from the evidence and the scale of the study. It should be based on your data to draw some implications but not too far away from it.

L246: Recently

L250: delete "kind of", not a very formal written expression. Also "playing"

L263: subjects? What subjects? Be clear. And also, it is noted that the authors references many clinical microbiome publications, not particularly broiler associated, or at least food producing animal related ones. It may not be a good practice as the biological functions between human and broilers could be vastly different, thus hindering the relevance of the study to the animal production. Suggest to look for more relevant papers.

Recommend to read one of the newly published paper about chicken gut microbiota which is relevant to your study.

<https://doi.org/10.1111/1751-7915.13841>

L273-295: This paragraph mainly talks about the correlation between certain taxa and body weight, e.g. fat diet and obesity. I suggest to reduce the content or summarise it to make it short, due to similar function of taxa listed.

L296-320: Refer to the previous comment, if the transcriptome analysis is removed, this paragraph should be considered too.

L467, L600: Correct the reference format.

Fig 3: Suggest to re-arrange the order of taxa as 4W, 14W and 16W on the y axis for ease of comprehension of data.

Fig 4A: Generally 3D illustration isn't usually a good choice, because it isn't as practical and easy as it would be, particularly when many data points are clustered together, which in this case data points of HW and LW at the lower PCoA dimensions are closely overlapped, making it hard to justify their correlation.

Reviewer #3 (Comments for the Author):

This manuscript was interesting to read. In the first half of the paper, the authors have profiled gut microbiota and its development from 4 to 16 wks. of age. The later second half looked at the association of microbiome with high and low-weight broilers. However, the major drawback of this study is that it is correlational lacking any validation study. The authors need to address the following issues before publication.

Comments:

Please mention upfront in the result section that the dynamic changes in gut microbiota are cumulative data from high and low-weight broilers.

How have authors claimed that immune level, glucose metabolism, and lipid metabolism is affected by mineral and vitamin metabolism that affects body weight? This is a very bold statement in the absence of supporting data. Experiments are not done to show which factor affects which one?

Were the samples measured repeatedly? In this case, account for the repeated measure.

Fig 1A and Fig S1 indicate that alpha diversity is affected by sample read numbers. Plot number of quality reads against weeks. If so, account for the read numbers for alpha and beta diversity analysis.

Are the analyses for Fig1-3 combined of both high and low weight broilers?

Figure 4: Show all the analysis in a longitudinal fashion, not as a single time point style.

Can a consortium six bacterial genera that were positively associated with body weight impact weight gain in mice/broilers? This experiment can be quickly performed to validate the results.

The positive association of selenium metabolism with body weight is an exciting observation. Please measure the level of selenium in the gut and system. This finding can be very significant to the field.

L20; Update with recent references.

L39; Make sentences more concise.

L57; A lower case. Provide updated data and references too.

L86; Are alpha diversity and body wt. significantly correlated? Show stats than just mentioning the downward trend.

L115-23; Arrange the order of Figure 4 according to the manuscript.

L122; Stats?

L145; There are not eleven genera in the figure.

L134-45; Also show the results from Lefse analysis.

L165; Describe briefly what is level 3 pathway.

L170; Which nine pathways?

L193; Please also use hallmark gene sets for GSEA. <http://www.gsea-msigdb.org/gsea/msigdb/collections.jsp>

L236; Not entirely true.

L241; Which data? Please reference to the figure and/or table for all the following and where possible in discussion section.

L249; Which data?

L260; Which data?

L270; Which data?

L309-10; Which data?

Reviewer #4 (Comments for the Author):

Due to high-correlation of each subject to itself in longitudinal data, could authors indicate how they accounted for repeated measures.

How PERMANOVA on alpha and beta-diversity analysis performed was not clear. Is it a univariate analysis or multivariate analysis? Were any confounding factors fed into the model? how repeated measures were taken into account

Staff Comments:

Preparing Revision Guidelines

To submit your modified manuscript, log onto the eJP submission site at <https://spectrum.msubmit.net/cgi-bin/main.plex>. Go to Author Tasks and click the appropriate manuscript title to begin the revision process. The information that you entered when you

first submitted the paper will be displayed. Please update the information as necessary. Here are a few examples of required updates that authors must address:

Please return the manuscript within 60 days; if you cannot complete the modification within this time period, please contact me. If you do not wish to modify the manuscript and prefer to submit it to another journal, please notify me of your decision immediately so that the manuscript may be formally withdrawn from consideration by Microbiology Spectrum.

Dynamic changes in the gut microbial community and function during broiler growth

This manuscript was interesting to read. In the first half of the paper, the authors have profiled gut microbiota and its development from 4 to 16 wks. of age. The later second half looked at the association of microbiome with high and low-weight broilers. However, the major drawback of this study is that it is correlational lacking any validation study. The authors need to address the following issues before publication.

Comments:

Please mention upfront in the result section that the dynamic changes in gut microbiota are cumulative data from high and low-weight broilers.

How have authors claimed that immune level, glucose metabolism, and lipid metabolism is affected by mineral and vitamin metabolism that affects body weight? This is a very bold statement in the absence of supporting data. Experiments are not done to show which factor affects which one?

Were the samples measured repeatedly? In this case, account for the repeated measure. Fig 1A and Fig S1 indicate that alpha diversity is affected by sample read numbers. Plot number of quality reads against weeks. If so, account for the read numbers for alpha and beta diversity analysis.

Are the analyses for Fig1-3 combined of both high and low weight broilers?

Figure 4: Show all the analysis in a longitudinal fashion, not as a single time point style.

Can a consortium six bacterial genera that were positively associated with body weight impact weight gain in mice/broilers? This experiment can be quickly performed to validate the results.

The positive association of selenium metabolism with body weight is an exciting observation. Please measure the level of selenium in the gut and system. This finding can be very significant to the field.

L20; Update with recent references.

L39; Make sentences more concise.

L57; A lower case. Provide updated data and references too.

L86; Are alpha diversity and body wt. significantly correlated? Show stats than just mentioning the downward trend.

L115-23; Arrange the order of Figure 4 according to the manuscript.

L122; Stats?

L145; There are not eleven genera in the figure.

L134-45; Also show the results from Lefse analysis.

L165; Describe briefly what is level 3 pathway.

L170; Which nine pathways?

L193; Please also use hallmark gene sets for GSEA. <http://www.gsea-msigdb.org/gsea/msigdb/collections.jsp>

L236; Not entirely true.

L241; Which data? Please reference to the figure and/or table for all the following and where possible in discussion section.

L249; Which data?

L260; Which data?

L270; Which data?

L309-10; Which data?

Dear Editor John Atack,

Thank you very much for processing our manuscript “*Dynamic changes in the gut microbial community and function during broiler growth*” (ID: Spectrum01005-22) for possible publication in *Microbiology Spectrum*. We are very grateful to the editor and the reviewers for their helpful suggestions and positive comments and for giving us an opportunity to revise the manuscript. We have carefully considered the comments and have made corrections that we hope will meet with your approval. For all modifications, please refer to the section marked in red in the revised manuscript. Our point-by-point responses to the reviewer's comments are marked in red and presented following.

Response to comments of the Reviewers:

Reviewer #1 (Comments for the Author):

The authors examined the fecal microbiome of broiler chickens on multiple timepoints (4-16 weeks) using 16S amplicon sequencing. They show a difference in the microbial composition during the different timepoints, including a decrease in alpha diversity, beta-diversity based clustering and specific bacteria that change. They also try to see the association between body weight and the microbial composition, albeit less convincingly (see specific comments below). They also look at the predicted gene content using picrust2, and again look for age/weight related pathways. Finally, they performed transcriptomics in 6 chickens from a single timepoint, and try to identify host genes associated with body weight. The cohort is large (10 chickens * 6 time points), focuses on a less-studied population (age 4-16 weeks). and results seem clean of major artifacts. However, some of the statistical analysis is less well supported and/or well described (i.e. application of FDR to some analysis, use of t-test, some observations do not have any statistical support - see specific comments below), and some results are over-stated/interpreted.

Response: We appreciate the constructive feedback and sincerely thank the reviewer for the thoughtful and supportive recommendations. Our responses to the comments from reviewer are given below. We sincerely thank the reviewer for the useful suggestions, which are of great help to improve the quality of this manuscript. In addition, combined with the opinions of reviewers and the actual situation of this study, we use the microbial data and transcriptome data of the ninth week to explore the differences between HW and LW. This has changed some results. We have made changes in the manuscript. Please check them.

Specific comments:

Methods:

1. What chickens were sampled at each time point? If I understand correctly out of 120 chickens, 10 were randomly chosen at each time point? If so, are there chickens that participated in more than 1 timepoint? What is the reason for this design? Please clarify in the methods section.

Response: Thank you for your precious comments and advices. We performed a random sampling

from 120 male Tianfu broiler chickens in 6 time point (week 4/5/8/9/14/16) without putting them back, 10 chickens per sampling. Each chicken was sampled only once at all time points. We believed that this random sampling method can better reflect the situation of larger groups. As suggested, the sampling method was described in more detail in the manuscript (Lines 295-297). "Ten individuals from 120 broilers in week 4, 5, 8, 9, 14 and 16 were randomly selected without putting back to collect fecal samples by swab."

2. were the 16S PCR primers barcoded? how long were the barcodes? there is "and index codes were added." is this the stage where the barcodes were added? If so, how were they added?

Response: 16S PCR primers were not barcodes. The length of each barcode in this study was 12nt (e.g. R110116: ATGTCA, CAACTA). And we provide the primer sequence and barcode sequence used in this study in **Table S13**. Yes, barcodes were added according to specificity.

3. was each chicken separately housed? please add to methods section.

Response: Thanks for your suggestion. Each chicken was separately housed. We have stated it in the manuscript (Lines 291-292).

"Broilers were subjected to the same feeding regimen and environment and were housed separately in cages and free access to commercial broiler fodder and water."

4. "We referred to the SILVA database to obtain the taxonomic composition of microorganisms." - how exactly was taxonomy assigned? using what software/parameters?

Response: Microbial classification was completed in QIIME2 analysis process, and default parameters (qiime feature-classifier classify-sklearn --i-reads --i-classifier --o-classification) were used. We have made a more detailed statement in the manuscript (Lines 328-330).

"We referred to the SILVA database to obtain the taxonomic composition of microorganisms in QIIME2 analysis process, and default parameters (qiime feature-classifier classify-sklearn --i-reads --i-classifier --o-classification) were used."

5. "Functional enrichment including GO terms and KEGG pathways of differentially expressed genes was performed by Metascape (88), and $p < 0.05$ was considered as significant." - I think this relates to the 16S picrust analysis? if so, please move to the relevant section in the methods.

Response: Thanks for your suggestion. This content is used to describe the functional enrichment of differentially expressed genes (DEGs) in transcriptome data analysis. We have moved it to the method section in the manuscript (Lines 362-364).

"Functional enrichment including GO terms and KEGG pathways of differentially expressed genes was performed by Metascape (64), and $p < 0.05$ was considered as significant."

Results:

6. "and increased from the initial average weight of 304.2 g in week 4 to 3, 105.8 g in week 16" is it supposed to be 3105.8? If so, please write without the space and the comma between "3" and "105.8"

Response: Thanks for your suggestion. Changes have been made as suggested (Lines 89-90).
"However, the body weight of broilers changed significantly and increased from the initial average weight of 304.2 g in week 4 to 3105.8 g in week 16 (Fig. 1C; Table S2)."

7. "These results showed that there were significant and dynamic changes in the gut microbial community of broilers after 4 weeks of age." - this sentence is not clear (what is "dynamic changes"?). Would recommend rephrasing or removing

Response: Thanks for your suggestion. We removed "dynamic" in the manuscript (Lines 94-95).
"These results showed that there were significant changes in the gut microbial community of broilers after 4 weeks of age."

8. "we correlated the average relative abundance of each time group to feeding time" - what does this mean? what is the "feeding time"? did you mean you identified bacteria correlated with the sample timepoint? how was correlation performed (pearson/spearman/other)? are the significant bacteria shown in figure 3A? what p-value threshold? was FDR correction applied?

Response: Yes, "feeding time" means "sampling time point". We calculated the Pearson correlation coefficients between bacteria and sampling time points at the genus level, with $p < 0.05$ and $FDR < 0.25$ being considered as significant correlation. Figure 3A shows the bacteria that are significantly related to the sampling time points. FDR correction was applied here. We have described it in more detail in the manuscript (Lines 334-337).

"The Pearson correlation coefficient between the average relative abundance of each genus in all groups and the sampling time points was calculated. $P < 0.05$ and $FDR < 0.25$ were considered to be the thresholds of significant correlation."

9. "indicating that the gut microbiota changed dramatically in week 14." I think "dramatically" is a bit of an overstatement. Is the week 14 change much bigger than the previous week changes? if not, maybe rephrase to "indicating that the gut microbiota continues to change even in week 14".

Response: Thanks for your valuable suggestion. We have modified it in the manuscript (Lines 117-118).

"There were 4 significantly different genera in week 14, indicating that the gut microbiota continues to change even in week 14."

10. "These results also proved that the gut flora changes dynamically during the growth of broilers." - "proved" is a bit strong. can change to something like "These results indicate" or "These results show".

Response: Thanks for your valuable suggestion. We have modified it in the manuscript (Lines 118-119).

"These results indicated that the gut flora changes dynamically during the growth of broilers."

11. "Analysis of gut microbial diversity and composition in high and low weight broilers" section:
How were the chickens divided into the high and low weight groups? What was the weight cutoff? was it the same in all time points? Also, why are there 23+25=48 chickens total in both groups combined and not 60? Please supply more details (maybe in the methods section).

Response: Thank you for your precious comments and advices. After consideration, we found that there may be some problems in the previous grouping method. In order to adapt the 16S data and transcriptome data of high and low weight grouping in this study, we used the data of the week 9 in this study to analyze the differences between HW and LW. For 16S rDNA data, the top 5 individuals with body weight were assigned as HW and the bottom 5 individuals were assigned as LW. We have revised the grouping method in manuscript (Lines 120-124).

"To obtain a more precise understanding of the differences in gut microbial diversity between high and low weight broilers, the samples with the highest and lowest body weight in week 9 were divided into high body weight (HW) (n = 5) and low body weight (LW) (n = 5) groups respectively."

12. "However, 3 dimensions principal co-ordinates analysis (3D-PCoA) based on weighted Bray-Curtis distances showed a trend towards separation (Fig. 4A)" I did not manage to see a trend in the PCoA. Do you have some statistic (such as permanova) to support your claim that the bacterial community differs between the high and low groups?

Response: As suggested, and combined with the opinions of other reviewers, we have used two-dimensional PCoA plot to show the results, and PERMANOVA test on HW and LW was performed. The result showed that their separation was not significant. We revised the statement in the manuscript (Lines 125-126).

"Although not significant ($p > 0.05$), PCoA based on weighted Bray-Curtis distances showed a trend towards separation (**Fig. 4A**)"

13. "These results illustrated that grouping was relatively reasonable." not clear to me what is the meaning of this sentence. What is a "reasonable grouping"? What supports your claim that this grouping is reasonable? Maybe delete this sentence or explain better.

Response: Thanks for your helpful suggestion. As you suggested, we have deleted this sentence.

14. "Firmicutes and Proteobacteria were relatively overrepresented in HW," - this whole paragraph has a lot of statements without statistical support. Since we are comparing two finite groups the mean frequency of each bacteria will always be different (i.e. not exactly the same). However, in order to claim that a specific bacteria has a different mean in the 2 populations (and not just in the random subset we are looking at), we need to use some statistical test to reject the null hypothesis (of similar mean in the 2 original groups). Please supply some statistical test results (i.e. lefse or mann-whitney with FDR correction) or alternatively remove the higher/lower/over-represented/under-represented parts of this paragraph.

Response: Thanks for your helpful suggestion. We conducted a Mann-Whitney test on the relative abundance of phyla in HW and LW, but they were not significant. We have removed the higher/lower/over-represented/under-represented parts in this paragraph.

15. "Microbes associated with body weight" - was correlation performed on all 60 samples or a subset? If on all 60 samples, how do you control/separate the effect of aging (i.e. older chickens weigh more) from the effect of specific bacteria (that supposedly lead to increased weight)? If no control is applied, please add a note in the results and discussion sections mentioning that the association may be indirect, due to both weight and bacterial composition being associated with age.

Response: Thank you for your reminder. Correlation analysis was performed with 48 individuals who had accurate weight records (the weight records of some samples have been lost). We have not set up a control group in this analysis, so we have explained it in the discussion section according to your tips in the manuscript (Lines 267-269).

"At the same time, it should be noted that the genera significantly related to weight identified by Spearman correlation in this study may have an indirect impact on weight, due to both weight and bacterial composition being associated with age."

16. "Microbes associated with body weight" - why did you use pearson correlation and not spearman? Spearman searches for linear correlation, which does not have to be the case with bacteria/weight relations, whereas pearson only looks for monotonic relation (which to me sounds more biologically probable). Note that it is totally ok to use pearson if you prefer, and the results are valid. But maybe spearman will provide more significant bacteria? Note that this is just a recommendation - you do not have to change the manuscript if you prefer pearson.

Response: Thanks for your helpful suggestion. As suggested, we have used Spearman correlation for analysis and found more statistically significant bacteria ($p < 0.01$ and $FDR < 0.01$). We have revised the relevant statements in the manuscript (Lines 138-143).

"To explore the microbes associated with broiler body weight, we calculated the Spearman correlation coefficients between the relative abundances of top 100 (from all 60 samples) genera and individual body weight. 3 genera with significant positive correlation with individual weight were selected ($p < 0.01$, $FDR < 0.01$ and $r > 0.6$), and 13 genera with significant negative correlation with individual weight were selected ($p < 0.01$, $FDR < 0.01$ and $r < -0.6$) (**Table S7**)."

17. "Microbes associated with body weight" - was the association performed at the genus level? or the ASV level? please clarify. If it was performed at the ASV level, please provide the specific ASV sequences in table S6

Response: Thanks for your reminder. The correlation analysis was performed at the genus level. We have explained it in the manuscript (Lines 138-140).

"To explore the microbes associated with broiler body weight, we calculated the Spearman correlation coefficients between the relative abundances of top 100 (from all 60 samples) genera and individual body weight."

18. "Microbes associated with body weight" - was FDR control used for the significant weight-associated bacteria? If not, please apply FDR rather than just using p-values (as you test 100 different bacteria, we would expect 5 to have $p < 0.05$ even if the null hypothesis holds for all bacteria). Note that even if no bacteria show significant association after FDR, it is fine to write so and does not reduce the power of the manuscript.

Response: Thanks for your helpful suggestion. We have used FDR control for the significant weight-associated bacteria in **Table S7**. And we took $p < 0.01$ and $FDR < 0.01$ as the significance threshold. And we have made corresponding modifications to the statement in the manuscript (Lines 140-143).

"3 genera with significant positive correlation with individual weight were selected ($p < 0.01$, $FDR < 0.01$ and $r > 0.6$), and 13 genera with significant negative correlation with individual weight were selected ($p < 0.01$, $FDR < 0.01$ and $r < -0.6$) (**Table S7**)."

19. "STAMP analysis with differential relative abundances between HW and LW revealed potential effects" - There is no description in the methods section of how STAMP was used for the HW/LW groups analysis (it only states STAMP was used for picrust2 analysis). Please supply details how STAMP was used - what groups were compared, was there control for age, what statistical test was used, was there multiple hypothesis control, what group of features was compared (100 most common or all, genus or ASV level or all phylogenetic levels) etc.

Response: Thanks for your helpful suggestion. Combined with the comments of other reviewers, we have used LEfSe to analyze the relative abundance of all genera between HW and LW in week 9. And $p < 0.05$ and $LDA > 3$ were used as thresholds. The corresponding method description has been stated in the manuscript (Lines 330-332).

"Linear discriminant analysis (LDA) coupled with linear discriminant analysis effect size (LEfSe) was used to identify differentially abundant taxa among various time groups, and between HW and LW at the genus level (55)."

20. "7647 Kos" - need to capitalize to O - "7647 KOs"

Response: As suggested, we have modified it in the revised manuscript (Lines 156-157).

"We acquired a total of 7647 KOs and identified 298 pathways in 60 fecal samples."

21. "Constrained principal component analysis showed that the function of microbiota in the gut also changed significantly in different time groups" - please give details regarding the software implementation and parameters analyzed using constrained-PCA in the methods section.

Response: Thanks for your helpful suggestion. Relevant details have been supplemented in the manuscript (Lines 341-343).

"On the ImageGP platform (58), weighted Bray-Curtis distances were calculated based on the relative abundance of PICURSt2 predicted KEGG pathways, and constrained principal co-ordinate analysis (CPCoA) of all samples was performed."

22. "The pathway annotation revealed that 49.13% of the total enriched genes were enriched in pathways related to metabolism" - how was the enriched genes identified? using STAMP? what groups were compared? what statistical test was used? what parameters? please provide details.

Response: "Enriched genes" refer to all 16S rRNA genes identified by PICRUSt2 and annotated to KEGG functional pathways. We have made more specific statements in the manuscript (Lines 162-165).

"The pathway annotation revealed that 49.13% of the total identified 16S rRNA genes were enriched in pathways related to metabolism, followed by pathways related to genetic information processing, environmental information processing and unclassified, accounting for 22.81%, 15.12% and 6.39% of the total enriched genes, separately."

23. "In addition, membrane transport, carbohydrate metabolism, replication and repair, amino acid metabolism, energy metabolism, translation were identified as the main functional pathways in the secondary classification" - what is the secondary classification? what is the difference from the analysis above? do you mean different kegg levels? please provide details for this analysis as well.

Response: "Secondary classification" refers to level 2 of KEGG pathways. We have list all 3 levels predicted KEGG functional pathway descriptions in Table S8.

24. "Furthermore, 5 pathways were significantly associated with individual weight among the top 100 level 3 pathways, they are selenocompound metabolism, chaperones and folding catalysts, streptomycin biosynthesis, thiamine metabolism and carbon fixation pathways in prokaryotes, respectively ($p < 0.05$)" - how were these pathways identified? pearson correlation with weight? please specify. Also, was multiple hypothesis correction (FDR) used (since the test was performed on 100 pathways, we would expect 5 with $p < 0.05$ even if the null hypothesis is correct for all pathways)? If not, please use FDR or another multiple hypothesis control method (ok to use $FDR < 0.1$ or even $FDR < 0.25$).

Response: Thank you for your precious comments and advices. After consideration, we have utilized Spearman correlation for correlation analysis, and FDR control was performed with $p < 0.01$ and $FDR < 0.1$ as thresholds. The corresponding statement has been revised in the manuscript (Lines 171-174).

"Furthermore, 14 KEGG pathways were significantly associated with individual weight among the top 100 (from all 60 samples) level 3 pathways ($p < 0.01$ and $FDR < 0.1$), they were mainly distributed in a variety of metabolic pathways and some information processing pathways (Table S9)."

25. "Through Welch's T test conducted with STAMP software, we found 4 differential pathways between HW and LW " - what groups of chickens did you use for the test? the same 23+25 as used in the 16S analysis? please provide details. Also, did you control for age as an effector of the pathway abundance? (see comment above regarding effect of age on weight)

Response: We have sorted the 10 samples in week 9 by weight and divided them into two groups: HW (n=5) and LW (n=5), and then Mann-Whitney test was performed between HW and LW by STAMP software. We have revised the corresponding statement in the manuscript (Lines 174-175). "Through Mann-Whitney test conducted with STAMP software in week 9, we found 5 differential pathways between HW and LW ($p < 0.05$ but $FDR > 0.25$) (Fig. 6B)."

26. "Through Welch's T test conducted with STAMP software, we found 4 differential pathways between HW and LW ($p < 0.05$)" - i do not think t-test is appropriate here, as it is a parametric test that assumes normality. Can you show the pathway abundance follows a somewhat normal distribution rather than zero-inflated? Otherwise, please use Mann-Whitney instead. Also, did you use multiple-hypothesis correction (FDR)? If not, please apply FDR, as you are testing multiple pathways (can use a threshold of 0.1 or even 0.25). If nothing turns out significant, it is still ok and does not reduce the importance of your manuscript.

Response: As you suggested, we have changed to Mann-Whitney test and carried out FDR correction. We have revised the corresponding statement in the manuscript (Lines 174-175). "Through Mann-Whitney test conducted with STAMP software in week 9, we found 5 differential pathways between HW and LW ($p < 0.05$ but $FDR > 0.25$) (Fig. 6B)."

27. "The results showed that the biological duplicate samples in the same group have relatively high correlation coefficients with each other" - what are the "biological duplicates"? Do you mean samples from the same weight group (LW/HW)? Please clarify.

Response: Thanks for your reminder. "Biological duplicates" refers to samples from the same weight group (LW/HW). However, after calculating the average Spearman correlation coefficients, the results did not support this statement, so we have removed this sentence in the manuscript.

28. "The results showed that the biological duplicate samples in the same group have relatively high correlation coefficients with each other" - I did not see a strong difference between the within-group vs. between group sample correlations (fig S3). Do you have some calculation to show the within group correlation is significantly higher than the between-group correlation? If not, would recommend to remove this sentence.

Response: Thank you for your suggestion. After calculating the average Spearman correlation coefficients, the results did not support this statement, so we have removed this sentence in the manuscript.

29. "Furthermore, the Principal components analysis (PCA) results showed that there was a certain degree of variance between HW and LW (Fig. 7A)." - What does "a certain degree of variance between HW and LW" mean? please clarify. Hard for me to see a difference between the 2 groups based on Fig7A. Do you have some statistic (for example permanova) to show that there is a separation between the HW and LW groups? otherwise, please delete this sentence and the following sentence (" These results indicated that there were differences in gene expression in

cecum of broilers with different body weights.")

Response: Thank you for your precious comments and advices. We have carried out the PERMANOVA test, but the result is not significant ($p > 0.05$). We have modified "Furthermore, the Principal components analysis (PCA) results showed that there was a certain degree of variance between HW and LW (Fig. 7A)." to "In addition, the results of Principal component analysis (PCA) showed that there was no significant difference (PERMANOVA, $p > 0.05$) in the overall expression level between HW and LW (Fig. 7A)" in the manuscript (Lines 190-192). And we removed "These results indicated that there were differences in gene expression in cecum of broilers with different body weights".

"In addition, the results of Principal component analysis (PCA) showed that there was no significant difference (PERMANOVA, $p > 0.05$) in the overall expression level between HW and LW (Fig. 7A)"

Discussion:

30. "In this study, gut microbial diversity decreased and stabilized with broilers growth" - What supports the claim that the diversity "stabilized" with broilers growth? Please clarify or remove the "stabilized" part.

Response: Thank you for your suggestion. We have removed the "and stabilized " in the manuscript (Lines 233-234).

"In this study, gut microbial diversity decreased with broilers growth, perhaps due to flora competition and host immune modulation."

31. "which illustrated to some extent that the gut microbiota may be affected by multiple factors which from real environment and embodied the cooperation and competition of flora". I did not understand this sentence.

Response: We are sorry that we did not describe clearly. Previously we speculated that the drastic change of *Fusobacteria* at the phylum level and *Fusobacterium* at the genus level at week 8 was due to environmental factors and the competition of flora. Now we revised it to "The relative abundance of *Fusobacteria* at the phylum level and *Fusobacterium* at the genus level sharply increased in week 8, and then returned in week 9, which suggested the competition and self-regulation of gut flora" in the manuscript (Lines 234-236).

"The relative abundance of *Fusobacteria* at the phylum level and *Fusobacterium* at the genus level sharply increased in week 8 (44.25%), and returned in week 9 (4.3%), which suggested the competition and self-regulation of gut flora."

32. "We found that HW had higher alpha diversity than LW, which is different from the study in laying hens" - as this result was not significant (or even close to significant), it could be due to chance as the null hypothesis is not rejected with high confidence. Please write that the results were not significant, or remove.

Response: Thank you for your suggestion. Since we are now only concerned about the differences in

intestinal microorganisms between high and low weight broilers in week 9, we have modified "We found that HW had higher alpha diversity than LW, which is different from the study in laying hens" to "We found that HW had a lower alpha diversity, although not significant ($p > 0.05$), which is consistent with findings in 300 days old laying hens" in the manuscript (Lines 240-242).

"We found that HW had a lower alpha diversity, although not significant ($p > 0.05$), which is consistent with findings in 300 days old laying hens (38)."

33. "Minerals is considered micronutrients" - correct to "Minerals are considered micronutrients"

Response: Thanks for your helpful comment. Because of the change of the analysis results, the discussion about minerals has been deleted from the manuscript.

34. "indicating that microorganisms in high-weight broilers consumed more dietary selenium" - i think this is a bit unsubstantiated. Since picrust2 predictions are at the gene level, it only indicates that there are more bacteria with the potential for selenium metabolism. The hypothesis that high-weight broilers consumed more dietary selenium is a possible, but not the only explanation.

Response: Thank you for your valuable suggestions. Because of the change of the analysis results, the discussion about minerals has been deleted from the manuscript. We hope the more in-depth discussion in our revised manuscript is ok.

Figures:

35. fig 1B: was rarefaction applied before calculating Faiths PD? if so, to what depth? If not, please rarefy prior to calculating Faith PD, as it is a binary metric, which may be affected by read depth.

Response: Thank you for your valuable suggestions. Yes, rarefaction was applied before calculating Faiths PD. The depth was 15000 (Figure S1).

36. fig 2B: I am a bit confused as the legend contains many (34) colors, but the bars in the plot seem to contain a much smaller number of colors?

Response: Sorry for that, we have redrawn the stacked bar plot of top20 genera (from all samples), and applied colors that are easier distinguished to represent different genera (Fig 2B).

37. fig 3: "with feeding time" - what is "feeding time" - do you mean the sample week (4/5/8/9/14/16)? please clarify.

Response: Thanks for your reminder. "Feeding time" refers to different sampling time points (4/5/8/9/14/16 week). We have modified "feeding time" to "different sampling time points (4/5/8/9/14/16 week)" in the revised manuscript (Lines 572-574).

"(A) The genera that their average relative abundance within each time point was significantly correlated with different sampling time points (4/5/8/9/14/16 week) (Pearson, $p < 0.05$ and FDR < 0.25)."

38. fig 5: please supply more details in the legend (i.e. what software/statistical test was used, what groups were compared etc.)

Response: Thank you for your comment. As suggested, more details in the legend were provided. We have corrected the legend of Figure 5 in the manuscript (Lines 587-588).

"Genera with significant difference between HW and LW identified by LEfSe analysis in week 9 (LDA > 3 and $p < 0.05$)."

39. fig 6B: please supply more details in the legend - how was significance calculated? what test/parameters?

Response: Thank you for your valuable suggestion. We have modified the legend of figure 6B in the manuscript (Lines 590-591).

"The significantly differential pathways between HW and LW (Mann-Whitney, $p < 0.05$ but FDR > 0.25)."

40. fig S6: please specify what each sampleid means (i'm assuming 1_X is the LW group, 2_X is the HW)? Please clarify in the legend if so (or another explanation)

Response: Thank you for your comment. We used Spearman correlation to measure the correlation between transcriptome samples and regenerated the heatmap (Figure S3). According to your suggestion, we have modified the legend of Figure S3 and S6.

"Figure S3. Spearman correlation heatmap of transcriptome samples. Blue and red represent low and high correlation, respectively. R92_X represents the LW group and R91_X represents the HW."

Data availability:

41. The accession provided in the text is good. If possible, please add the per-chicken weight to the metadata, and specify also the per-sample timepoint (i.e. age weeks) in addition to collection date.

Response: Thanks for your suggestion. The corresponding information has been added to Table S12.

Reviewer #2 (Public repository details (Required)):

The 16S microbiota dataset needs to be uploaded to the public repository.

Response: Thank you for your valuable suggestion. The 16S rRNA gene sequencing data and transcriptome sequencing data involved in this study have been released. We have provided the available SRA BioProject ID: PRJNA779029 in the manuscript (Lines 368-370) to facilitate the review of the data.

"Data availability. Original datasets are available in a publicly accessible repository: The original contributions presented in the study are publicly available. This data can be found here: [NCBI SRA (<https://www.ncbi.nlm.nih.gov/sra>); BioProject ID: PRJNA779029]."

Reviewer #2 (Comments for the Author):

The authors presented a study on the broiler gut microbiota in terms of its changes over the rearing period from 4 weeks to 16 weeks of age. The study offers insights into the composition of gut microbiota between high and low weight broilers and predicts the functions related to the weight gain/loss. It could provide useful information into the broiler production.

Response: We appreciate the constructive feedback and sincerely thank the reviewer for the thoughtful and supportive recommendations. Our responses to the comments from reviewer are given below. We sincerely thank the reviewer for the useful suggestions, which are of great help to improve the quality of this manuscript. In addition, combined with the opinions of reviewers and the actual situation of this study, we used the microbial data and transcriptome data of the ninth week to explore the differences between HW and LW. This has changed some results. We have revised them in the manuscript. Please check them.

42. Despite some minor grammar issues, the manuscript is reasonably well written. It is suggested to shorten some content in the discussion section. The discussion seems to be a bit of wordy.

Response: We deeply appreciate the reviewer's suggestion. As suggested, we corrected some grammar issues and simplified the content of the discussion in the manuscript.

43. L44: why "has" is crossed over?

Response: Thank you for your reminder. This is a writing mistake. We have modified the "~~has~~" to "has" in the manuscript (Line 44-45).

"In addition, accumulating evidence has shown that the gut microbiota affects the growth and development of the host."

44. L45: Read the journal guideline about how to do in-text citation. Similar issues noted throughout the manuscript, please correct them accordingly.

Response: Thank you for your precious comments. We have modified the in-text citation problems in the manuscript.

45. L51: what management are you referring here? Animal welfare, feed conversion ratio, etc.?

Response: "Management" here refers to various management measures during broiler husbandry, such as controlling ambient temperature, feed feeding volume, etc. After consideration, we feel that this sentence is inappropriate. Therefore, we deleted the sentence "the focus of poultry breeding industry is to optimize economic benefits under Limited breeding and management" in the manuscript.

46. L53: His -> Their. It's "Fredrik et al.", not "Fredrik".

Response: Thank you for your suggestion. We have modified "His" to "The" in the manuscript (Lines 52-53).

"The study revealed that the gut microbiota is a necessary condition for the body weight gain of the

host."

47. L77: As the way you present is the Results section first then the Methods, I suggest to provide a brief summary of what the samples were, the time points, etc. so it's easier to understand what you're assessing.

Response: Thank you for your suggestion. We have added a more detailed sample description at the start of the results section. Specifically, we revised "a total of 60 fecal samples collected from six time points were subjected to the microbial diversity study through a 16S rDNA high-throughput sequencing approach" to "a total of 60 fecal samples (10 samples per time point) collected from six time points (4/5/8/9/14/16 week) were subjected to the microbial diversity study through a 16S rRNA gene high-throughput sequencing approach" in the manuscript (Lines 76-80).

"In order to comprehensively understand the dynamics of gut microbial community in broiler during growth process, a total of 60 fecal samples (10 samples per time point) collected from six time points (4/5/8/9/14/16 week) were subjected to the microbial diversity study through a 16S rRNA gene high-throughput sequencing approach."

48. L89: You should have the full term spelled out when they are referred to for the first time, e.g. PCoA.

Response: Thank you for your helpful suggestion. We have modified "PCoA analysis" to "principal co-ordinates analysis (PCoA)" in the manuscript (Lines 91-92).

"Significant separation of gut microbiota from different time groups was observed using principal co-ordinates analysis (PCoA)."

49. L91: Suggest changing the expression to "except samples of Week 4 and 5, and Week 9 and 16", or "except between Week 4 vs. Week 5, and Week 9 vs. Week 16", to avoid the confusion.

Response: Thank you for your comment, which has improved the quality of our manuscript. We have changed "except between week 4 and week 5 and week 9 and week 16" to "except samples of Week 4 and 5, and Week 9 and 16" in the manuscript (Lines 92-94).

"By using PERMANOVA pairwise comparison, we confirmed that there were significant differences in the beta diversity of gut microbiota except samples of Week 4 and 5, and Week 9 and 16 ($p < 0.05$) (Table S4)."

50. L96: It is important to provide the version number of SILVA that you accessed for 16S analysis.

Response: Thank you for your suggestion. We have provided version number "(132_99 release)" in the manuscript (Lines 98-99).

"Compared with the SILVA database (132_99 release), total of 36 phyla, 89 classes, 216 orders, 390 families, 852 genera and 1, 176 species were identified among 6 groups."

51. L97: Top 10 from each time point or all together? Same with Top 33.

Response: "Top 10" and "Top 33" here refer to the top 10 phyla of all samples and the top 33 genera of

all samples. After consideration, we now show the top 10 phyla of all samples and the top 20 genera of all samples. We have changed "(top 10)" to "(top 10 from all samples)" and "(top 33)" to "(top 20 from all samples)" in the manuscript (Lines 99-101).

"The most abundant phyla (top 10 from all samples) and genera (top 20 from all samples) with high average relative abundance were displayed using the stacked bar chart (Fig. 2)."

52. L102: Due to the colour similarity of the stacked bar chart making it hard to tell the accurate proportion of taxa, i would suggest providing the detail of the relative abundances when certain organisms are listed at various time points. For instance, the proportions of *Lactobacillus* over some weeks collection are hard to compare. Same with other taxa you have listed here. We don't know to what extend the high level or low level was in the dataset.

Response: Thank you for your helpful suggestion. For the information displayed by the stacked bar chart, we provide the corresponding Table S5 to show the relative abundance at different time points in detail. And we used colors that are easier to distinguish to represent different phyla and genera in Fig 2A and Fig 2B.

53. L108: Similar to the previous comment, details are needed to examine the change of microbiota over time. Just stating significant with a p value nor the exact proportion isn't enough for assessment. Same applies to the following genera changes in Week 14. What were they? How much were they changed at which time points?

Response: We are sorry for the inappropriate expression. This sentence is to describe that in Figure 3A, the genera whose abundance were significantly negatively correlated with the sampling time points were mainly concentrated in the four phyla *Firmicutes*, *Proteobacteria*, *Bacteroidetes* and *Actinobacteria*. The corresponding abundance information and trend can be obtained from Fig. 3A. We have modified the corresponding statement in the manuscript (Lines 110-114).

"To elaborate more nuanced changes, we screened the genera whose abundance was significantly related to the sampling time points ($p < 0.05$ and $FDR < 0.25$). At the phylum level, we observed that the genera with reduced abundance were mainly concentrated in Firmicutes, Proteobacteria, Bacteroidetes and Actinobacteria. Moreover, the genera with increased abundance were mainly concentrated in Firmicutes and Proteobacteria (Fig. 3A)."

The following description of LEfSe analysis results is to explain that microbial changes have been occurring. According to the requirements of Reviewer 1, we have modified the corresponding statement in the manuscript (Lines 115-118).

"Based on LEfSe results, we observed a total of 8 significantly different genera from week 4, 14 and 16 ($LDA > 4$ and $p < 0.05$) (Fig. 3B). There were 4 significantly different genera in week 14, indicating that the gut microbiota continues to change even in week 14."

54. On a personal note, it is a problem publishing microbiome papers when detailed information is missing from all the colourful charts in the results section. The data visualisation is fancy and

colourful, but it doesn't provide the robust assessment for the purpose of scientific evaluation. People won't be able to tell the exact abundance of taxa from all the colourful bars, let alone if some researchers are colour blind, which isn't a rare case.

Response: We are sorry for this problem. However, in this study, there are many bacteria with low abundance. Although we use colors that are easier to distinguish, their difficulty in reading is still a problem. Therefore, we used Table S5 and Table S6 to provide detailed information, hoping that their existence will help to understand the manuscript.

55. L121: Honestly it's hard to tell the separation from the 3D illustration. Unless you can rotate it to find a better viewing point to show the separation, it's not a good way to present data in most cases. A PCoA with clustering is a better option.

Response: Thank you for your suggestion. As suggested, we have used PCoA analysis to show the results in Fig 4A, and we used PERMANOVA test to judge whether the separation was significant. The result showed that $p > 0.05$. Therefore, we have revised the statement in the manuscript (Lines 125-126).

"Although not significant ($p > 0.05$), PCoA based on weighted Bray-Curtis distances showed a trend towards separation (Fig. 4A)."

56. L127: Again, how much did they present in LW? You should provide a good comparison between HW and LW in terms of some phyla if you want to state their different abundance in two sample groups. Listing them in HW without giving their abundance in LW doesn't help with the evaluation.

Response: Thank you for your suggestion. Combined with the comments of the Reviewer 1, we have statistically tested the abundance of these phyla in HW and LW. However, they were not significant, so we have removed the relevant statements in the manuscript.

57. L130: From the taxa below, I assume 43.34% is from HW. But it would be better to have it more specific and clearer.

Response: Thank you for your suggestion. We have described the abundance of these genera in HW and LW more clearly. For example, "*Romboutsia* (HW: 5.51%; LW: 9.27%)" in the manuscript (Lines 132-137).

"At the genus level, *Lactobacillus* was the most abundant genus in two groups, with relative abundances of 62.9% from HW and 62.62% from LW respectively. Besides, *Romboutsia* (HW: 5.51%; LW: 9.27%), *Candidatus Arthromitus* (HW: 7.75%; LW: 1.19%), *Fusobacterium* (HW: 5.58%; LW: 3.02%), *Bacteroides* (HW: 3.36%; LW: 4.37%) and *Escherichia-Shigella* (HW: 1.85%; LW: 1.3%) also had a high abundance occupancy in HW and LW, respectively (Fig. 4E; Table S6)."

58. L140: What were they averaged abundance over time?

Response: Thank you for your suggestion. We have listed the relative abundance of corresponding genera in the manuscript (Lines 144-147).

"Among these body weight-related genera, we further screened out 4 genera with high relative abundance (relative abundance > 1%), and they were *Romboutsia* (10.24%), *Corynebacterium* 1 (1.66%), *Gallibacterium* (1.42%) and *Faecalibacterium* (1.04%)."

59. L145: It is important to highlight their roles in terms of positive or negative impact on the weight gain/loss.

Response: Thank you for your suggestion. We have deleted this sentence and discussed the relevant results in the manuscript (Lines 246-258).

"*Bilophila* was associated with obesity in mice induced by high-fat diet (40), but it was also found to be significantly negatively correlated with broiler weight (41). As a potential beneficial bacterium, *Akkermansia* was more abundant in the better fed efficacy group hens (42). *Akkermansia muciniphila* can prevent body weight loss by alleviating intestinal mucosal damage caused by dysentery in chickens (43). And dietary supplementation with sophorolipid could significantly increase the feed utilization and increase the abundance of *Akkermansia muciniphila* in chicks (44). The abundance of *Candidatus Arthromitus* decreased during the period of broiler infected with necrotic enteritis (45), indicating that *Candidatus Arthromitus* may be a beneficial bacterium for broilers. In this study, we found that *Bilophila*, *Epulopiscium*, *Akkermansia*, and *Candidatus Arthromitus* were more abundant in HW. Combined with the results of us and other researchers above, we speculated that *Akkermansia* and *Candidatus Arthromitus* are conducive to the healthy weight gain of broilers, and *Bilophila* needs to be further studied on the relationship with the weight gain of broilers. Through Spearman correlation analysis, we screened 16 genera and 14 pathways that might have an effect on broiler body weight, respectively. Among the 16 genera, *Romboutsia*, *Gallibacterium* and *Methanobrevibacter* were significantly positively correlated with body weight. *Methanobrevibacter* was found to promote abdominal fat deposition in broilers (37). Dietary supplementation with *Bacillus subtilis* improved growth performance of broilers which may be related to the increased abundance of *Romboutsia* (46). However, the severe respiratory infection caused by *Gallibacterium anatis* will seriously affect the laying of hens (47). Combined with the results listed above, we should conduct more in-depth studies on the mode of action of genera significantly associated with body weight."

60. L165: were

Response: We have changed "are" to "were" in the manuscript (Lines 171-174).

"Furthermore, 14 KEGG pathways were significantly associated with individual weight among the top 100 (from all 60 samples) level 3 pathways ($p < 0.01$ and $FDR < 0.1$), they were mainly distributed in a variety of metabolic pathways and some information processing pathways (Table S9)."

61. L173: Although the results of transcriptome of broiler cecum yielded some interesting indications, the sample size and the result isn't sufficient or significant to draw any meaningful conclusion.

Focus on the microbiota to avoid distraction of the study purpose.

Response: Thank you for your valuable suggestions. Here we performed the analysis of transcriptome

data to aid in illustrating the potential mode of influence of the gut microbiota on broiler body weight, only as a supplement to functional analysis of the gut microbiota. After consideration, we still decided to keep the related statement, but in the discussion section we gave it less attention in the manuscript (Lines 273-275).

"Transcriptome analysis showed that the differentially expressed genes between HW and LW were related to immunity (*LOC107051274* and *ENSGALG00000048383*) and energy metabolism (*ART4*, *MT-CYB* and *MT-ND4*)."

62. L210: And -> and, same in L213. Minor grammar mistakes.

Response: Thank you for your reminder. We have modified them in the manuscript (Lines 220-224).

"This is a supplement to previous research, and, overall, increased relative abundances of Firmicutes and decreased relative abundances of Proteobacteria. These indicate that large changes in the gut microbiota and its composition of broilers after 4 weeks of age, and appears to portend a common changing trend of gut microbiota across multiple species as well as shared dominant phyla."

63. L215: If choosing to remove the transcriptome analysis, this paragraph should be altered. Focus on the microbiota of your data and its possible implication.

Response: Thank you for your proposal. We retained the transcriptome analysis, but due to the change of the analysis results, we still made changes to the discussion part. And we paid more attention to microbiological analysis, while reducing the discussion of transcriptome data results.

64. L228-231: This is a very long sentence, with too many "which". Break it up into a few sentences with a logic flow is better.

Response: Thank you for your valuable advice. Combined with the opinions of other reviewers, we revised the statement in the manuscript (Lines 234-236).

"The relative abundance of Fusobacteria at the phylum level and Fusobacterium at the genus level sharply increased in week 8, and returned in week 9, which suggested the competition and self-regulation of gut flora."

65. L237-238: This speculation is a bit of too stretched from the evidence and the scale of the study. It should be based on your data to draw some implications but not too far away from it.

Response: Thank you for your valuable advice. We have removed this conjecture related statement from the manuscript (Lines 240-242).

"We found that HW had a lower alpha diversity, although not significant ($p > 0.05$), which is consistent with findings in 300 days old laying hens (38)."

66. L246: Recently

Response: Thank you for your valuable comments. After the correction of the manuscript, we removed this sentence from the revised manuscript, indicating that this error no longer exists.

67. L250: delete "kind of", not a very formal written expression. Also "playing"

Response: Changes have been made as suggested.

68. L263: subjects? What subjects? Be clear. And also, it is noted that the authors references many clinical microbiome publications, not particularly broiler associated, or at least food producing animal related ones. It may not be a good practice as the biological functions between human and broilers could be vastly different, thus hindering the relevance of the study to the animal production. Suggest to look for more relevant papers. Recommend to read one of the newly published paper about chicken gut microbiota which is relevant to your study.

<https://doi.org/10.1111/1751-7915.13841>

Response: Thank you for your valuable comments. After the correction of the manuscript, we removed this sentence from the revised manuscript. In addition, we have removed references that are not suitable for this study, and referred to more research papers related to broiler intestinal microorganisms. The research you mentioned was also cited in the manuscript (Lines 58-59). "Fecal transplantation could also improve the growth performance and liver fat metabolism of broilers by reshaping the intestinal microbiota (18)."

69. L273-295: This paragraph mainly talks about the correlation between certain taxa and body weight, e.g. fat diet and obesity. I suggest to reduce the content or summarise it to make it short, due to similar function of taxa listed.

Response: Thank you for your valuable comments. After the correction of the manuscript, we removed this sentence from the revised manuscript, indicating that this errors no longer exist. At the same time, according to the corrected results, we discussed them in the manuscript (Lines 259-267). "Through Spearman correlation analysis, we screened 16 genera and 14 pathways that might have an effect on broiler body weight, respectively. Among the 16 genera, *Romboutsia*, *Gallibacterium* and *Methanobrevibacter* were significantly positively correlated with body weight. *Methanobrevibacter* was found to promote abdominal fat deposition in broilers (37). Dietary supplementation with *Bacillus subtilis* improved growth performance of broilers which may be related to the increased abundance of *Romboutsia* (46). However, the severe respiratory infection caused by *Gallibacterium anatis* will seriously affect the laying of hens (47). Combined with the results listed above, we should conduct more in-depth studies on the mode of action of genera significantly associated with body weight."

70. L296-320: Refer to the previous comment, if the transcriptome analysis is removed, this paragraph should be considered too.

Response: Thank you for your proposal. We retained the transcriptome analysis, but due to the change of the analysis results, we made further changes to the discussion part. And we paid more attention to microbiological analysis, while reducing the discussion of transcriptome data results.

71. L467, L600: Correct the reference format.

Response: Thank you for your precious comments. We have modified the citation problems in the manuscript (Lines 403-405). The reference at Lines 600 have been removed, so this reference problem no longer exists.

"9. Yatao X, Saeed M, Kamboh AA, Arain MA, Ahmad F, Suheryani I, Abd El-Hack ME, Alagawany M, Shah QA, Chao S. 2018. The potentially beneficial effects of supplementation with hesperidin in poultry diets. *World's Poultry Science Journal* 74:265-276."

72. Fig 3: Suggest to re-arrange the order of taxa as 4W, 14W and 16W on the y axis for ease of comprehension of data.

Response: Thank you for your suggestion. We have changed the order in Figure 3.

73. Fig 4A: Generally, 3D illustration isn't usually a good choice, because it isn't as practical and easy as it would be, particularly when many data points are clustered together, which in this case data points of HW and LW at the lower PCoA dimensions are closely overlapped, making it hard to justify their correlation.

Response: Thank you for your suggestion. We have used 2-dimensional PCoA plot to show the results in Figure 4A.

Reviewer #3 (Comments for the Author):

This manuscript was interesting to read. In the first half of the paper, the authors have profiled gut microbiota and its development from 4 to 16 wks. of age. The later second half looked at the association of microbiome with high and low-weight broilers. However, the major drawback of this study is that it is correlational lacking any validation study. The authors need to address the following issues before publication.

Response: We appreciate the constructive feedback and sincerely thank the reviewer for thoughtful and supportive suggestions. We are doing some experiments mentioned in your comments, but we may need more time to produce substantive results because we are not good at microbial culture and experiments. Our response to the comments of the reviewer is as follows. In addition, combined with the opinions of reviewers and the actual situation of this study, we only use the microbial data and transcriptome data of the ninth week to explore the differences between HW and LW. This has changed some results. We have made relative changes in the manuscript. Please check them.

Comments:

1. Please mention upfront in the result section that the dynamic changes in gut microbiota are cumulative data from high and low-weight broilers.

Response: Thank you for your valuable comments. After the correction of the manuscript, we used 10 samples in the ninth week to explore the microbial differences between HW and LW.

2. How have authors claimed that immune level, glucose metabolism, and lipid metabolism is

affected by mineral and vitamin metabolism that affects body weight? This is a very bold statement in the absence of supporting data. Experiments are not done to show which factor affects which one?

Response: Thank you for your comments. We made an inappropriate statement about this speculation. After the correction of the manuscript, we speculated that microorganisms may affect the immune level and energy metabolism level of broilers through their own carbohydrate metabolism and lipid metabolism, and then affect body weight. In this speculation, we emphasized that this was only a possibility.

3. Were the samples measured repeatedly? In this case, account for the repeated measure.

Response: We are sorry for the lack of clarity. There was no repeated measurement in this study, and we have explained it in the method part of the manuscript (Lines 295-297).

"Ten individuals from 120 broilers in week 4, 5, 8, 9, 14 and 16 were randomly selected without putting back to collect fecal samples by swab"

4. Fig 1A and Fig S1 indicate that alpha diversity is affected by sample read numbers. Plot number of quality reads against weeks. If so, account for the read numbers for alpha and beta diversity analysis.

Response: Thank you for your suggestion. Figure S1 showed that with the increase of sequences, the number of detected OTUs first increased and gradually remained stable, indicating that we need to choose the appropriate sequence number for microbial data analysis. In this study, when sequence number=15000, the number of detected OTUs basically reached the maximum. We chose sequence number=15000 for subsequent licensing, ensuring the reliability of microbial data analysis. We have added relevant statements to the manuscript (Lines 83-85).

"We selected the sequence number = 15000 for subsequent analysis. In this case, the rarefaction curves were close to saturation, indicating that the sequencing depth was adequate in all the samples (Fig. S1). "

5. Are the analyses for Fig1-3 combined of both high and low weight broilers?

Response: The analysis for Figures 1-3 contains all 60 16S rRNA gene sequencing samples.

6. Figure 4: Show all the analysis in a longitudinal fashion, not as a single time point style.

Response: Thank you for your advice. Combined with the opinions of other reviewers and considering the adaptability of microbial data and transcriptome data, we decided to use the microbial data and transcriptome data of the ninth week for the difference analysis between HW and LW.

7. Can a consortium six bacterial genera that were positively associated with body weight impact weight gain in mice/broilers? This experiment can be quickly performed to validate the results.

Response: Thank you for your proposal. We are in the process of isolating and culturing chicken derived intestinal microorganisms, but because this work is not our advantage and has certain

difficulties, we have not yet obtained the target strain. However, we will continue to isolate and culture in order to more accurately study their impact on the weight of broilers.

8. The positive association of selenium metabolism with body weight is an exciting observation. Please measure the level of selenium in the gut and system. This finding can be very significant to the field.

Response: Thank you very much for your suggestion. After the correction of the manuscript, we did not find the association between selenium metabolism and body weight. The relationship between microbial selenium metabolism and broiler weight may need more research to explain.

9. L20; Update with recent references.

Response: As suggested, we have updated some references.

10. L39; Make sentences more concise.

Response: As suggested, we modified the sentence in our revised manuscript (Lines 39-40).

"Some studies shown that gut microorganisms can affect host metabolism (2-6), disease resistance (7, 8) and development."

11. L57; A lower case. Provide updated data and references too.

Response: Thank you for your reminder. We have changed "A" to "a" and updated the reference in the manuscript (Lines 56-58 and 423-426).

"More importantly, a colonization study of the mouse gut microbiota demonstrated that obesity can transmit by specific colonization of the gut microbiota (17)."

"17.Ridaura VK, Faith JJ, Rey FE, Cheng J, Duncan AE, Kau AL, Griffin NW, Lombard V, Henrissat B, Bain JR, Muehlbauer MJ, Ilkayeva O, Semenkovich CF, Funai K, Hayashi DK, Lyle BJ, Martini MC, Ursell LK, Clemente JC, Van Treuren W, Walters WA, Knight R, Newgard CB, Heath AC, Gordon JI. 2013. Gut microbiota from twins discordant for obesity modulate metabolism in mice. Science 341:1241214."

12. L86; Are alpha diversity and body wt. significantly correlated? Show stats than just mentioning the downward trend.

Response: Thank you for your reminder. What we want to express here is that with the extension of time, alpha diversity shows a downward trend. According to your suggestion, we have calculated the Spearman correlation coefficient between the average alpha diversity and sampling time points, and found that alpha diversity was negatively correlated with sampling time ($p = 0.09$ and $r = -0.74$).

Therefore, we modified the statement in the manuscript (Lines 87-88).

"And with the growth of broilers, the alpha diversity showed a downward trend (Pearson, $r = -0.74$ and $p = 0.09$)"

13. L115-23; Arrange the order of Figure 4 according to the manuscript.

Response: As suggested, we have modified the order of relevant statements in the manuscript to make the order of Figure 4 consistent with the manuscript (Lines 120-137).

"To obtain a more precise understanding of the differences in gut microbial diversity between high and low weight broilers, the five samples with the highest body weight in week 9 were divided into high body weight (HW) (n = 5) and the five samples with the lowest body weight in week 9 into low body weight (LW) (n = 5). The weight difference between HW and LW was significant (Fig. S6A and B), indicating that the grouping was reliable. Although not significant ($p > 0.05$), PCoA based on weighted Bray-Curtis distances showed a trend towards separation (Fig. 4A). Then, we used Faith's PD and Shannon indices to visualize the microbiota diversity in two groups. High weight broilers had a lower diversity, but the difference was not significant (Mann-Whitney, $p > 0.05$) (Fig. 4B and C).

Next, we explored changes in the microbial composition of the two groups. At the phylum level, Firmicutes, Bacteroidetes, Proteobacteria and Fusobacteria were dominant phyla in both HW and LW (relative abundance $> 2\%$) (Fig. 4D; Table S6). At the genus level, Lactobacillus was the most abundant genus in two groups, with relative abundances of 62.9% from HW and 62.62% from LW respectively. Besides, Romboutsia (HW: 5.51%; LW: 9.27%), Candidatus Arthromitus (HW: 7.75%; LW: 1.19%), Fusobacterium (HW: 5.58%; LW: 3.02%), Bacteroides (HW: 3.36%; LW: 4.37%) and Escherichia-Shigella (HW: 1.85%; LW: 1.3%) also had a high abundance occupancy in HW and LW, respectively (Fig. 4E; Table S6)."

14. L122; Stats?

Response: According to your suggestion, we have carried out PERMANOVA test on PCoA analysis in the manuscript (Lines 125-126).

"Although not significant ($p > 0.05$), PCoA based on weighted Bray-Curtis distances showed a trend towards separation (Fig. 4A)."

15. L145; There are not eleven genera in the figure.

Response: We are sorry for the lack of a clear description. The 11 genera here refer to the sum of 6 genera significantly related to body weight plus 5 genera with significant differences between HW and LW. After the correction of the manuscript, this part has been removed in the manuscript.

16. L134-45; Also show the results from Lefse analysis.

Response: Combined with your suggestion, we have used LefSe to analyze the genera of the difference between HW and LW in the manuscript (Lines 148-151).

"LefSe analysis between HW and LW based on the relative abundance of all genera revealed potential effects of *Bilophila*, *Epulopiscium*, *Akkermansia*, *Candidatus Arthromitus*, *Ruminococcaceae* UCG-013 and *Turicibacter* on the body weight of broilers ($p < 0.05$ and LDA Score > 3) (Fig. 5)."

17. L165; Describe briefly what is level 3 pathway.

Response: Level 3 pathways refer to the third level description in the KEGG pathways structure. You can see the pathways descriptions of different levels in Table S8.

18. L170; Which nine pathways?

Response: After the correction of the manuscript, this part has been modified in the revised manuscript (Lines 178-180).

"In total, these 19 pathways (Fig. 6B; Table S9) indicate that metabolic and information processing related pathways may be potential pathways for intestinal flora to affect broiler weight."

19. L193; Please also use hallmark gene sets for GSEA.

<http://www.gsea-msigdb.org/gsea/msigdb/collections.jsp>

Response: As suggested, rank-based gene set enrichment analysis (GSEA) was performed using clusterProfiler package and hallmark gene sets from MSigDB (www.gsea-msigdb.org/gsea/msigdb/), and the significant enrichment pathways with P_{adj} (FDR) < 0.05 and $|NES| > 1$ were identified. And we have added the corresponding content to the manuscript (Lines 200-204 and 364-367).

"Through gene set enrichment analysis (GSEA), 8 pathways were found to be enriched in two BW groups (Fig. 7D). 7 pathways (such as "unfolded protein response", "G2M checkpoint", "mTORC1 signaling") were enriched in LW (Fig. S5A-G), and "xenobiotic metabolism" was enriched in HW (Fig. S5H)."

"Rank-based gene set enrichment analysis (GSEA) was performed using clusterProfiler (v4.0.5) (65) package and hallmark gene sets from MSigDB (www.gsea-msigdb.org/gsea/msigdb/), and the significant enrichment pathways with P_{adj} (FDR) < 0.05 and $|NES| > 1$ were identified."

20. L236; Not entirely true.

Response: Thank you for your suggestion. We have revised this sentence in the manuscript (Lines 240-242).

"We found that HW had a lower alpha diversity, although not significant ($p > 0.05$), which is consistent with findings in 300 days old laying hens (38)."

21. L241; Which data? Please reference to the figure and/or table for all the following and where possible in discussion section.

Response: This information previously appeared in the attachment. After the correction of the manuscript, these statements no longer correspond to the results, so these contents have been removed from the manuscript.

22. L249; Which data?

Response: This information previously appeared in the attachment. After the correction of the manuscript, these statements no longer correspond to the results, so these contents have been removed from the manuscript.

23. L260; Which data?

Response: This information previously appeared in the attachment. After the correction of the manuscript, these statements no longer correspond to the results, so these contents have been removed from the manuscript.

24. L270; Which data?

Response: This information previously appeared in the attachment. After the correction of the manuscript, these statements no longer correspond to the results, so these contents have been removed from the manuscript.

25. L309-10; Which data?

Response: This information previously appeared in the attachment. After the correction of the manuscript, these statements no longer correspond to the results, so these contents have been removed from the manuscript.

Reviewer #4 (Comments for the Author):

We appreciate the constructive feedback and sincerely thank the reviewer for the thoughtful and supportive recommendations. Our responses to the comments from reviewer are given below.

1. Due to high-correlation of each subject to itself in longitudinal data, could authors indicate how they accounted for repeated measures.

Response: Thank you for your reminder. We are sorry for the lack of clarity. There was no repeated measurement in this study, and we have explained it in the method part of the manuscript (Lines 295-296).

"Ten individuals from 120 broilers in week 4, 5, 8, 9, 14 and 16 were randomly selected without putting back to collect fecal samples by swab"

2. How PERMANOVA on alpha and beta-diversity analysis performed was not clear. Is it a univariate analysis or multivariate analysis? Were any confounding factors fed into the model? how repeated measures were taken into account.

Response: We are sorry for the lack of a clear description of statistical testing methods. We used Kruskal Wallis H test to statistically test the alpha diversity between each two groups. And we use a univariate PERMANOVA tests to test the beta-diversity between each two groups. We have explained them in the manuscript (Lines 323-327). Except for the sampling timepoint, no other confounding factors were included in the model. There was no repeated measurement in this study, and we have explained it in the method part of the manuscript (Lines 295-296).

"For metrics of taxonomic diversity, we used Shannon and Faith's phylogenetic diversity (Faith's PD) (54) indices to compute the alpha diversity, and different time groups were compared using the Kruskal–Wallis H test. The comparisons of gut microbiota composition between time groups in the growth process were conducted by univariate permutational analysis of variance (PERMANOVA) for the Bray-Curtis distances and illustrated by principal co-ordinates analysis (PCoA)."

"Ten individuals from 120 broilers in week 4, 5, 8, 9, 14 and 16 were randomly selected without putting back to collect fecal samples by swab"

July 26, 2022

Prof. Diyan Li
Sichuan Agricultural University
College of Animal Science and Technology
Chengdu
China

Re: Spectrum01005-22R1 (Dynamic changes in the gut microbial community and function during broiler growth)

Dear Prof. Diyan Li:

Your manuscript has been accepted, and I am forwarding it to the ASM Journals Department for publication. You will be notified when your proofs are ready to be viewed.

Sincerely,

John Atack
Editor, Microbiology Spectrum

Journals Department
Supplemental Tables: Accept
Supplemental Figures: Accept